https://doi.org/10.1038/s42003-022-03498-3　　**OPEN**
# AMAISE: a machine learning approach to index-free sequence enrichment

Meera Krishnamoorthy[1], Piyush Ranjan[2], John R. Erb-Downward[2,3], Robert P. Dickson [2,3,4] & Jenna Wiens [1✉]

Metagenomics holds potential to improve clinical diagnostics of infectious diseases, but DNA from clinical specimens is often dominated by host-derived sequences. To address this, researchers employ host-depletion methods. Laboratory-based host-depletion methods, however, are costly in terms of time and effort, while computational host-depletion methods rely on memory-intensive reference index databases and struggle to accurately classify noisy sequence data. To solve these challenges, we propose an index-free tool, AMAISE (A Machine Learning Approach to Index-Free Sequence Enrichment). Applied to the task of separating host from microbial reads, AMAISE achieves over 98% accuracy. Applied prior to metagenomic classification, AMAISE results in a 14–18% decrease in memory usage compared to using metagenomic classification alone. Our results show that a reference-independent machine learning approach to host depletion allows for accurate and efficient sequence detection.

[1] Division of Computer Science and Engineering, Department of Electrical Engineering and Computer Science, University of Michigan, Ann Arbor, MI, USA. [2] Division of Pulmonary & Critical Care Medicine, Department of Medicine, University of Michigan, Ann Arbor, MI, USA. [3] Department of Microbiology and Immunology, University of Michigan, Ann Arbor, MI, USA. [4] Max Harry Weil Institute for Critical Care Research and Innovation, University of Michigan, Ann Arbor, MI, USA. ✉email: wiensj@umich.edu

Recent advances in sequencing technology have made metagenomics a promising tool for the clinical identification of pathogens in infectious diseases[1–3]. Given recent progress in the cost and speed of sequencing, the generation of sequencing data is no longer the major barrier to implementation. Instead, the primary hurdle is the computational challenge of pathogen identification amidst the copious host-derived sequences that dominate sequencing results from clinical specimens[1,4]. To improve the efficiency and accuracy of metagenomic classification of clinical specimens, prior investigators have employed laboratory-based host depletion steps (e.g., differential lysis) to deplete host DNA prior to sequencing[5]. These protocols require hours of hands-on work, increasing the time and cost of metagenomic analyses[5]. Computational depletion, an alternative to laboratory-based depletion, typically uses $k$-mer and sequence alignment-based methods to classify and remove host sequences[6]. These computational depletion approaches rely on large reference genome sets to identify and remove host sequences. This dependence on reference genomes results in inaccuracies when input reads do not match the reference genome due to sequencing inaccuracy (as is common among real-time long-read sequencing platforms) and real-world genomic variation. The large reference database requirement also necessitates large amounts of random access memory (RAM) or disk space, a major barrier to point-of-care translation. In contrast, machine learning models can classify DNA reads without relying on a reference database[7]. We hypothesized that this independence would lead to an improvement in memory efficiency and robustness to sequencing inaccuracies relative to $k$-mer and sequence alignment-based methods when used as a host depletion tool.

To that end, we present a reference-independent, machine learning-based host-depletion tool, AMAISE. We provide a comprehensive comparison of AMAISE and existing alignment and classification methods in their ability to discriminate host from microbial sequences. We also evaluate AMAISE as a pre-processing tool, applied before downstream microbial taxonomic classification. We perform this evaluation on a variety of test sets with differing host fractions, host species, and target microbiota. Read sets were not subject to pre-processing or quality control to test AMAISE's ability to handle noisy sequence data. We also perform this evaluation in two different computational environments: one with a large amount of RAM and powerful GPUs and one that mimics a laptop environment. We show that AMAISE achieves a comparable or higher accuracy at discriminating host reads from microbial reads than Kraken2[8], Centrifuge[9], and Minimap2[10]. Moreover, we show that AMAISE improves the accuracy and memory efficiency of the downstream classification by removing the reliance on host reference genomes.

## Results

**Overview**. AMAISE is based on a convolutional neural network (CNN) consisting of multiple convolutional layers followed by a single global average pooling layer (Fig. 1). Given a set of single-end reads, AMAISE outputs a classification label determining whether each sequence belongs to a host or a microbe (0 for microbe and 1 for host). AMAISE then stores all the microbial sequences in a file for downstream analysis (Fig. 1). AMAISE is lightweight, requiring only 0.003 GB of storage. This represents a 300 fold improvement in storage efficiency over common classification or alignment approaches.

We evaluated AMAISE (i) as a stand-alone host-depletion tool and (ii) in combination with existing metagenomic classifiers. As a stand-alone host-depletion tool, we first evaluated AMAISE against CNN-based approaches from past literature. We then evaluated AMAISE against several existing tools including

Kraken2, Centrifuge, and Minimap2. Applied before such techniques, AMAISE allows for efficient metagenomic classification by eliminating the reliance on host DNA reference genomes. Thus, we also evaluated AMAISE as a pre-processing method applied before a lightweight version of Kraken2 and Centrifuge in which the reference indices only contained microbial DNA.

We evaluated AMAISE on a laptop-like environment since this applies broadly in resource-constrained settings. However, since some of the existing approaches including Minimap2 and Kraken2 did not run successfully on the laptop-like environment, we also evaluated all methods on a server for a fair comparison.

We tested AMAISE on a variety of samples with Nanopore reads. We focused our evaluation on Nanopore reads because Nanopore sequencing currently produces the longest reads and is the only platform that produces real-time sequencing data, both useful for clinical diagnostics[1,11]. In our analyses, we first varied the percentage of reads of the sample that pertained to host. Specifically, we constructed 5 test sets with human host percentages (in percentage of number of reads) varying across 1%, 25%, 50%, 75%, and 99%, each with bacteria and fungi in the microbial fraction. We then varied the identity of the host and microbes in the sample by including 3 test sets with a pig host, mouse host, and viruses in the microbial fraction, respectively. The read sets were assembled from sequencing reads obtained from the NCBI SRA and did not have any post-sequencing modifications (e.g., modifications introduced by read cleaning and quality control procedures). These read sets were verified by alignment to their labeled taxonomy in SRA providing a consistent ground truth taxonomic assignment for each sequence.

Finally, to understand how our approach applies to real samples, we consider a test set of human oral metagenomic samples publicly available at the NCBI SRA (SRR11547004, SRR11547007). The best hits from BLAST[12] were used as ground truth labels for the real metagenomic test set.

**Comparison to other CNN-based approaches**. We compared AMAISE to CNN-based approaches from past literature[13–15] in terms of the ability to classify a synthetic test set with a human host percentage of 50% and bacteria and fungi in the microbial fraction. We focused on two weaknesses of previously proposed solutions: the assumption of a fixed input size and not training on data from both reference genomes and sequencing technologies. In contrast, AMAISE assumes a variable input size and was trained on sequences from a combination of reference genomes and Nanopore data. We isolate each of these aspects and determine how much each contributes to the speed and accuracy of AMAISE.

In contrast to solutions that pad sequences to the maximum length[14], AMAISE classified sequences more quickly (18 vs. 31 min) with no difference in sensitivity and specificity. While a CNN that truncated all sequences to be length 250 was 15 min faster than AMAISE (3 vs. 18 min)[13], AMAISE outperformed such an approach in terms of sensitivity by 3% (99% vs. 96%) and specificity by 8% (99% vs. 92%).

Training on only reference genomes[14] or only data from a sequencing technology[13,15], results in lower sensitivity (79% vs. 99%) and lower specificity (73% vs. 99%) respectively, compared to training on both reference genomes and Nanopore data.

**Host depletion comparison on server**. We evaluated AMAISE, Kraken2, Centrifuge, and Minimap2 in their ability to perform host depletion on a server with 64 AMD EPYC 7702 64-Core Processors (128 hyperthreaded cores), 256 GB of RAM, and 8 NVIDIA RTX 2080 GPUs[16]. As host-depletion tools, the reference indices of Kraken2, Centrifuge, and Minimap2 contain the

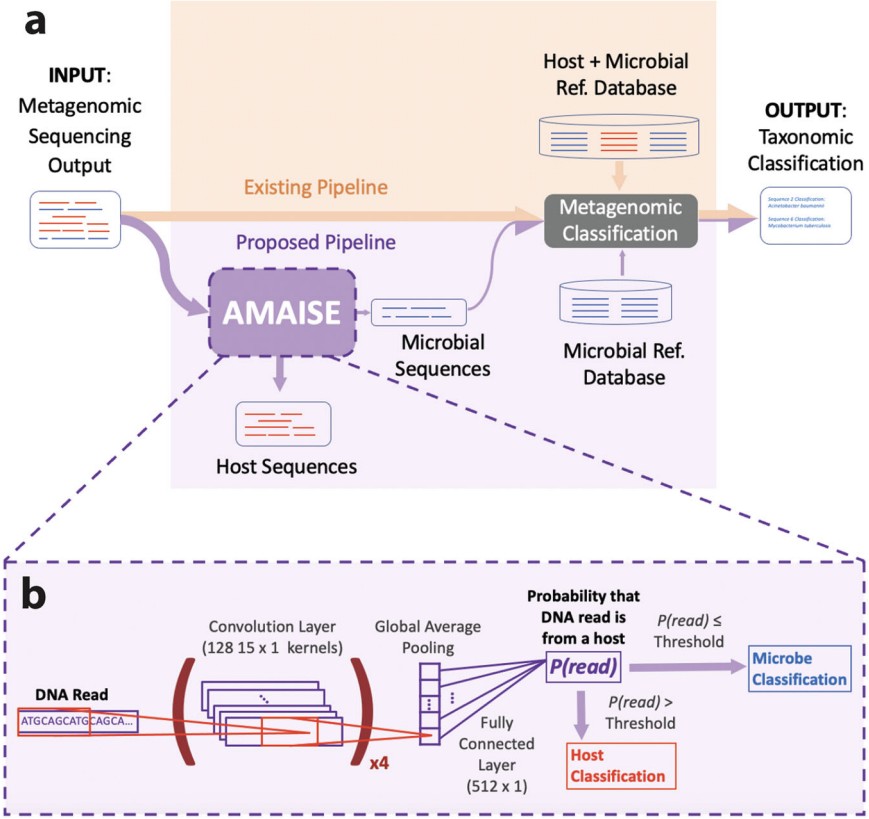

**Fig. 1 An overview of the AMAISE workflow. a** A comparison of the metagenomic classification pipeline with (purple) and without AMAISE (orange). In standard metagenomic community identification, reads are sent to a classification method that uses host and microbial references to assign a taxonomic label to each read. In our pipeline, we propose to import the reads into AMAISE, which then separates host from microbial reads. This smaller microbial fraction is sent to a classification method that only needs microbial references to assign taxonomy. **b** AMAISE is composed of four convolutional layers followed by a global average pooling layer. For each read input, AMAISE outputs a probability that the read is host. If the probability is higher than the predefined threshold, the read is considered to be from a host. Else, the read is considered to be from a microbe.

human genome assembly GRCh38.p13. We refer to these methods as Kraken2-H, Centrifuge-H, and Minimap2-H. We evaluated AMAISE against Kraken2-H, Centrifuge-H, and Minimap2-H in terms of classification time, peak memory usage, storage requirements, accuracy, sensitivity, and specificity (Fig. 2). Sensitivity represents the percentage of host sequences that are correctly identified as being host[17].

We chose to only include the human genome assembly GRCh38.p13 in the reference indices of Kraken2-H, Centrifuge-H, and Minimap2-H because we found that Minimap2-H's accuracy decreased when using a pan genome index. We discuss this further in Supplementary Note 1.

Across test sets with different host fractions, AMAISE achieved the highest sensitivity (>99%) compared to Kraken2-H, Centrifuge-H, and Minimap2-H. The existing method with the closest sensitivity, Centrifuge-H, achieved a maximum sensitivity of only 87.5%. In addition, AMAISE achieved comparable or higher specificity and accuracy (>98%) compared to existing methods. Furthermore, AMAISE ran in <20 min, on par with Kraken2-H and Minimap2-H, and between 10 and 50 min faster than Centrifuge-H. In addition, AMAISE's performance in terms of speed and accuracy remained robust across different host fractions, in sharp contrast to existing approaches (Fig. 2).

To test the ability of AMAISE when applied to real metagenomic data, we evaluated AMAISE on a test set of publicly available human oral metagenomic samples. On these data, AMAISE achieved a slightly higher specificity (91% vs. 85–90%) and comparable sensitivity and accuracy to other approaches. AMAISE ran in 6 min, approximately the same time it took to run Centrifuge but required less storage (0.003 vs. 1.5 GB) (Fig. 3a). We found that the real metagenomic data contained, on average, shorter reads than the synthetic data, with a median read length of around 900 bp compared to the 4000 bp median read length for the synthetic data (Fig. 3b). We also found that the host and microbial fractions in these data were more distinct than in the synthetic data. Specifically, the host and microbial fractions in the real metagenomic data had more unique 11-mers, 13-mers, 15-mers, and 17-mers than the host and microbial fractions in the synthetic data had (Fig. 3c).

To test the ability of AMAISE to classify samples with non-human sequences, we evaluated AMAISE on two samples, with a pig and mouse host, respectively, and confirmed that it had a high (>98%) accuracy on those sequences (see Supplementary Note 2, Supplementary Fig. S1). In addition, to test the ability of AMAISE to classify samples with microbes that are similar to humans, we evaluated AMAISE on a sample with viruses in the microbial fraction and confirmed that it had a comparable specificity (92%) on those sequences to Centrifuge (see Supplementary Note 3, Supplementary Fig. S2).

In terms of peak memory usage, on all datasets, AMAISE was on par with Kraken2-H requiring 3.0 GB of RAM and 1.6 GB of video RAM (VRAM) compared to Kraken2-H's requirement of 4 GB of RAM. However, AMAISE required less RAM than Minimap2-H. Finally, on all datasets, AMAISE required <0.3% of the storage required by the other baselines (Figs. 2 and 3).

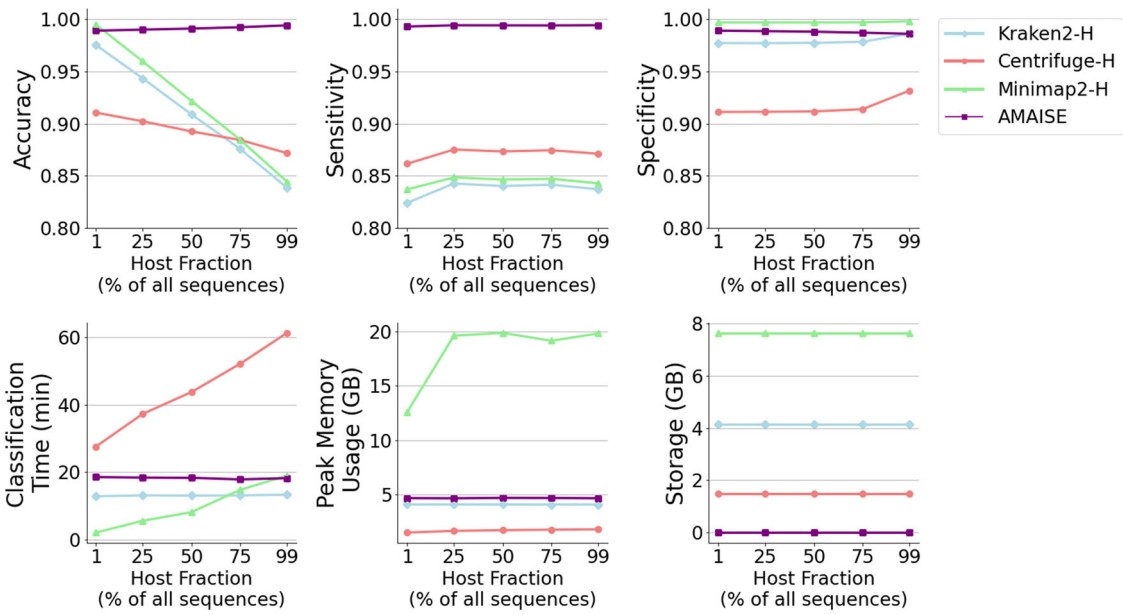

**Fig. 2 Performance of AMAISE, Kraken2-H, Centrifuge-H, and Minimap2-H across samples that varied in terms of host percentage.** Across samples that varied in terms of host percentage, the performance of AMAISE remained stable. AMAISE consistently achieved higher accuracy while requiring less storage and remaining competitive with respect to classification time and peak memory usage. Gains in accuracy and speed over Kraken2-H, Centrifuge-H, and Minimap2-H are largest when the fraction of host samples is above 25%.

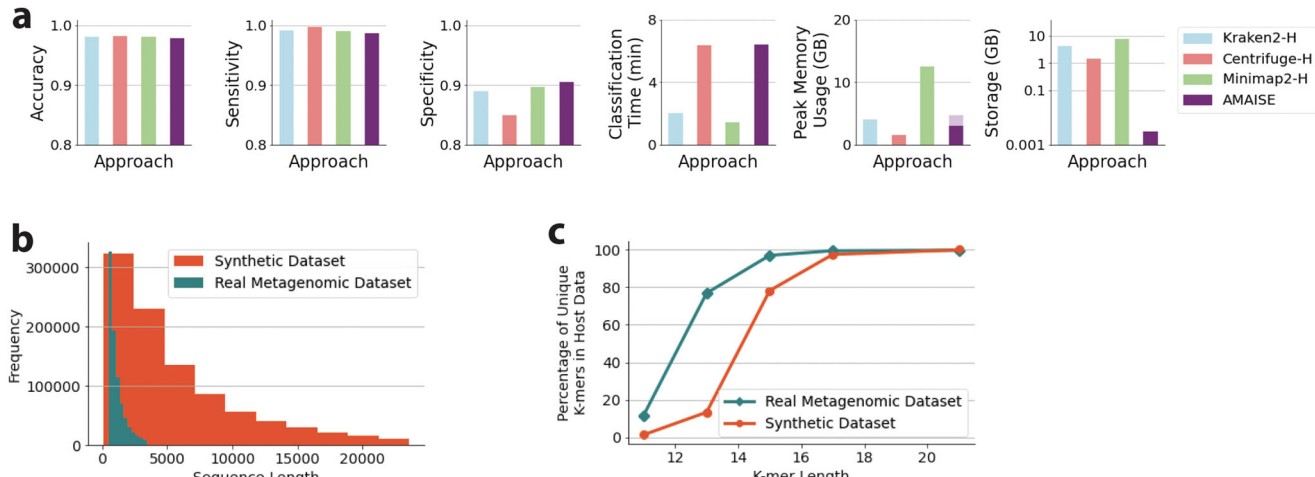

**Fig. 3 Performance of all methods on a real metagenomic test set; and comparison of the real metagenomic test set and synthetic test set. a** On real metagenomic data, AMAISE achieved a slightly higher specificity than other methods and remained competitive with respect to accuracy, sensitivity, classification time, and peak memory usage. **b** Length of real metagenomic test set vs. the synthetic test set. The real metagenomic data, on average, contained shorter sequences than the synthetic data. **c** Percentage of unique host *k*-mers in the real metagenomic test set compared to the synthetic test set. The real metagenomic data contains more unique 11-mers, 13-mers, 15-mers, and 17-mers in the host fraction compared to the synthetic data, which suggests that classifying the real metagenomic data is an easier task than classifying the synthetic data.

**Metagenomic classification pipeline comparison on server**. We assess AMAISE's effect on the performance of metagenomic classification methods, as described in Fig. 1, by comparing metagenomic classification performance of existing approaches with and without AMAISE. When used with AMAISE, existing approaches contain just microbial DNA in their reference indices and are referred to as *-M. Without AMAISE, existing approaches must contain both human and microbial DNA in their reference indices (and are referred to as *-HM) otherwise such approaches misclassify host sequences as microbial (see Supplementary Note 4, Supplementary Fig. S3).

Here, we focus our comparison on Centrifuge with (AMAISE + Centrifuge-M) and without AMAISE (Centrifuge-HM), since Centrifuge-HM outperformed Kraken2-HM on every metric except classification time (see Supplementary Note 5, Supplementary Fig. S4). We performed this evaluation on a server with 64 AMD EPYC 7702 64-Core Processors (128 hyper-threaded cores), 256 GB of RAM, and 8 NVIDIA RTX 2080 GPUs. We used the evaluation metrics of binary accuracy on the sequences labeled as host (whether each sequence that should have been labeled as a host was labeled as a host), multi-class accuracy on the sequences labeled as microbial (whether each

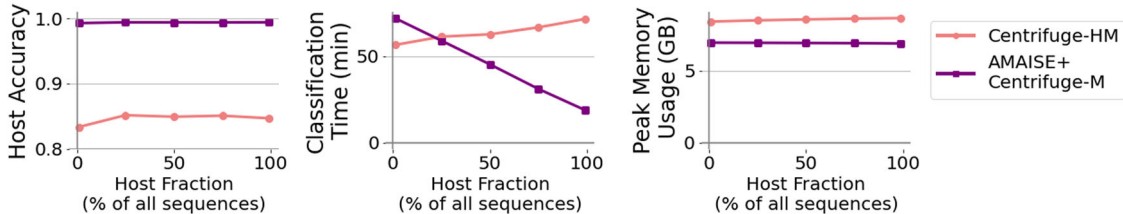

**Fig. 4 Performance of Centrifuge-HM and AMAISE + Centrifuge-M across samples that varied in terms of host percentage.** Across samples that varied in terms of host percentage, the pipeline that included AMAISE consistently achieved a higher host accuracy while requiring less peak memory usage and remaining competitive with respect to classification time. Notably, AMAISE + Centrifuge-M's classification time decreased as the percentage of host data in the test set increased, while Centrifuge-HM's classification time increased.

microbial sequence had an accurate species classification by Kraken2 or Centrifuge), classification time, peak memory usage (RAM + VRAM), and storage requirements.

Across test sets with different host fractions, AMAISE had the greatest impact on host sequence classification, improving the host accuracy of the Centrifuge-HM from 85% to over 99% (Fig. 4). Our findings were similar in terms of host accuracy with Kraken2 (see Supplementary Note 5, Supplementary Fig. S4).

AMAISE + Centrifuge-M's classification time decreased to below that of Centrifuge-HM's as the percentage of host data in the test set increased. Specifically, as the host data in the test set increased from 1% to 99%, AMAISE + Centrifuge-M's classification time decreased linearly from 70 min to <20 min. Centrifuge-HM's classification time had the opposite trend—as the host data in the test set increased from 1% to 99%, Centrifuge-HM's classification time increased linearly from 60 to 70 min (Fig. 4). Given that Kraken2 is optimized for speed, while the trend was similar, i.e., AMAISE + Kraken2-M took less time as the host fraction increased, including AMAISE in the pipeline did not lead to a decrease in classification time compared to Kraken2-HM (see Supplementary Note 5, Supplementary Fig. S4).

Across all test sets, AMAISE reduced the total storage requirements of Centrifuge-HM by 14% and the peak memory usage by 14%. We observed similar reductions in the context of Kraken2 (see Supplementary Note 5, Supplementary Fig. S4).

**Comparisons in a resource-constrained computational environment.** In a resource-limited computational environment designed to mimic a laptop (GCP VM), we evaluated AMAISE against Kraken2-H and Centrifuge-H in its ability to separate host from microbial data (Fig. 5a). In that environment, we also compared a pipeline including AMAISE and Centrifuge-M against a pipeline that just included Centrifuge-HM (Fig. 5b). The resource-limited computational environment had 4 vCPUs, 16 GB of RAM, 200 GB of boot disk storage, and 1 Tesla T4 GPU with 320 NVIDIA Turing Tensor Cores and 2560 NVIDIA CUDA cores[18]. Minimap2-H, Kraken2-M, and Kraken2-HM did not successfully run in our resource-limited computational environment.

The only evaluation metric that was affected across all methods by the change in computational environment was classification time (Fig. 5). AMAISE maintained the highest sensitivity with the lowest storage requirements compared to all other host-depletion approaches. Moreover, the pipelines including AMAISE still had the highest host and microbial accuracies with the lowest storage and memory requirements compared to the pipelines without AMAISE.

Across test sets with different host fractions, all methods were slower on the GCP VM than on the server. AMAISE's classification time increased by 59% on the GCP VM compared to the server, a higher percentage than other compared methods. While Centrifuge-H's classification time only increased by

15–30%, AMAISE on the GCP VM remained faster than Centrifuge-H on the GCP VM on test sets with host fractions above 25%. AMAISE + Centrifuge-M also remained faster than Centrifuge-HM on the GCP VM on test sets with host fractions above 25% (Fig. 5).

**Interpreting our machine learning model.** We examined the $k$-mers, specifically subsequences of length 15 learned in the first layer of AMAISE, that had the largest impact on the classification decision of AMAISE.

Using DeepLift, an algorithm that identifies the importance of input features on a machine learning model's classification, on a randomly selected subset of sequences in our test set, we determined the 15-mers that contributed to the classification decision[19–21]. The 15-mers that contributed most to a host classification label contained dimers and trimers that start with C or end with G. The 15-mers that most contributed to a microbial classification label contained at least one 'CG' and repeating dimers, and trimers of Ts (Fig. 6).

However, AMAISE does not just rely on 'CG' to make its classification decisions. Compared to a naive method that solely used the amount of 'CG' in a sequence to make its classification decision, AMAISE was more accurate (99% vs. 85%). Furthermore, we found that $k$-mer based and alignment-based methods rely on 'CG' content to a much larger extent. Specifically, Kraken2, Centrifuge, and Minimap2 had lower sensitivities than AMAISE on test sets with larger percentages of 'CG' (Fig. 7a). We found that the test sets with larger percentages of 'CG' had fewer unique $k$-mers in the host data compared to the test sets with smaller percentages (Fig. 7b). This suggests that classifying data with lower percentages of 'CG' is an easier task than classifying data with higher percentages of 'CG.'

**Discussion**
To leverage metagenomics in the clinical diagnosis of infectious diseases, classification pipelines must improve. Current computational approaches like Kraken2 and Centrifuge require the storage of host genomes, which is prohibitively costly in terms of storage and efficiency for current clinical environments. Moreover, given their reliance on matching input sequences to host genomes, these tools are often inaccurate when applied to noisy sequence data[6,8,9]. We created a host depletion tool, AMAISE, that eliminates this reliance on host genomes. In doing so, it increased Kraken2 and Centrifuge's accuracy and decreased their memory and storage requirements. Compared to existing host-depletion tools, AMAISE achieved the highest sensitivity with the lowest storage requirements, and comparable or better speed, specificity, and peak memory usage.

We developed AMAISE such that it could be applied before quality control techniques in research protocols and pipelines. Existing quality control measures are time-consuming and impractical for real-time applications (such as clinical

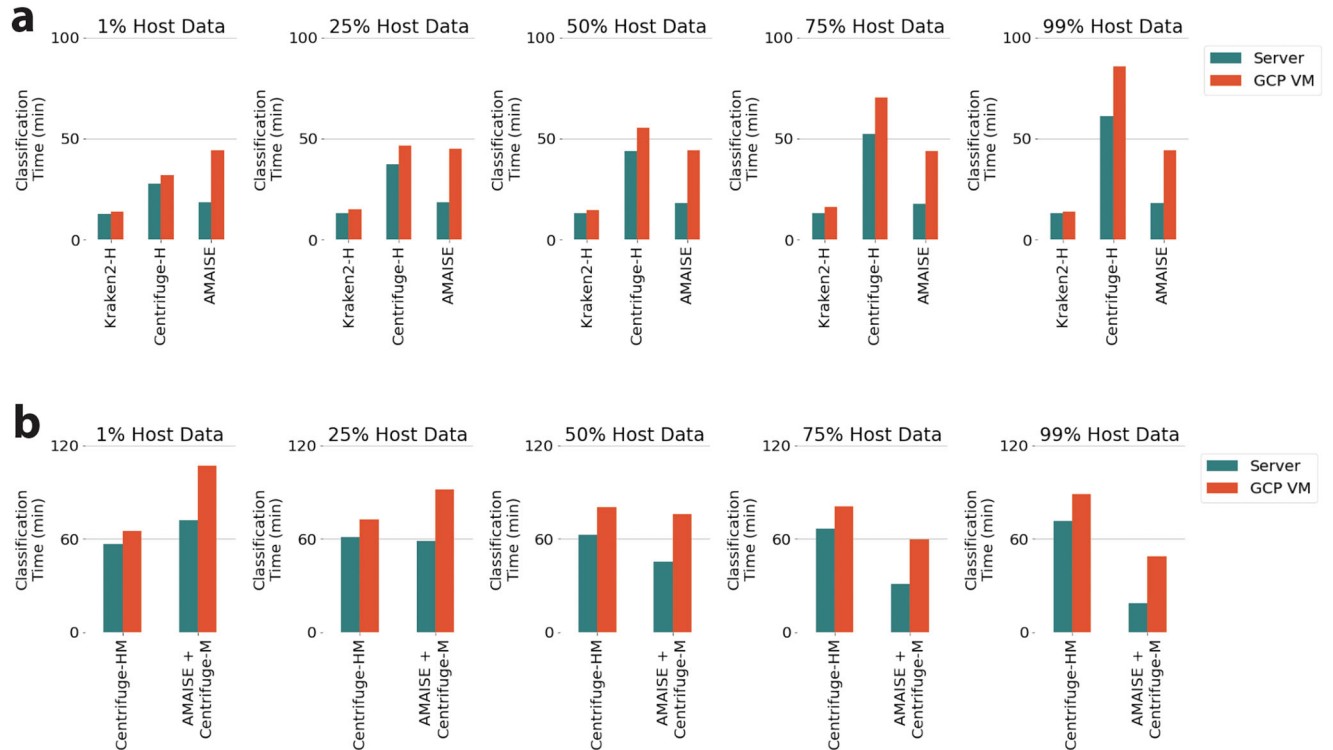

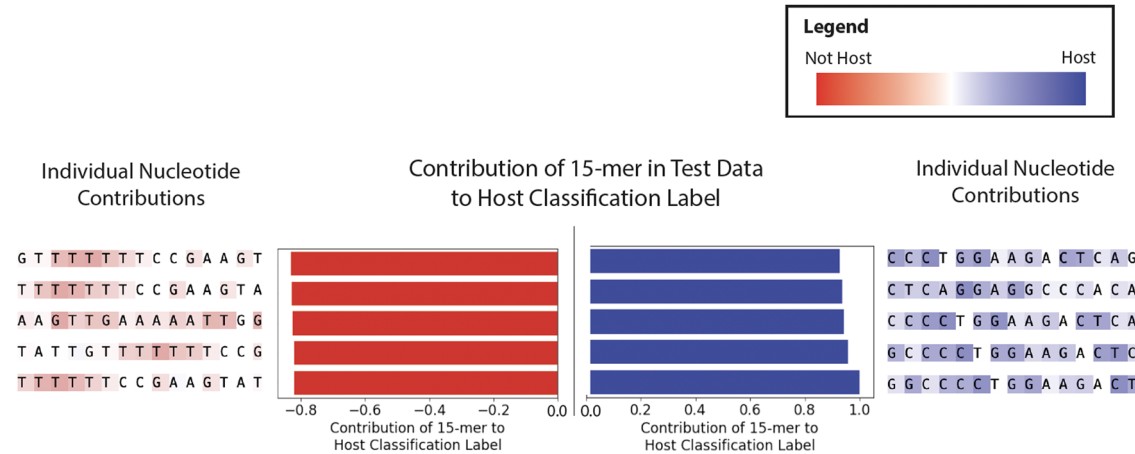

**Fig. 5 Classification time comparison of all methods in two different computational environments. a** AMAISE's speed is more affected by the change in computational environment than Kraken2-H and Centrifuge-H, but AMAISE is still faster than Centrifuge-H on test sets with host percentages exceeding 25%. **b** The pipeline including AMAISE and Centrifuge-M is faster than Centrifuge-HM on test sets with host percentages exceeding 25%.

**Fig. 6 15-mers that most and least contributed to AMAISE's host classification label.** The horizontal bar graphs represent the overall contribution of 15-mers to AMAISE's classification decision. The five 15-mers with the highest impact on the decision for each class, "not host" (or "microbe") (to the left) and "host" (to the right), are shown with their nucleotide sequences. Each nucleotide in the sequence is highlighted with a color/darkness indicating their individual impact towards a classification decision, red for microbial, blue for host classification, and dark for increased impact. Based on the individual impact of the nucleotides highlighted in red, we found that the 15-mers that most contributed to a microbial classification label contained at least one 'CG'.

diagnostics) especially when applied to a large number of host sequences. The ability of AMAISE to improve the accuracy and efficiency of existing methods without pre-processing via quality control and reference indices is a major advantage as we strive to bring sequencing to the bedside. This property is critical in resource-limited environments (e.g., clinical microbiology laboratories or point-of-care applications).

AMAISE is especially efficient and accurate at classifying noisy long-read data. Its efficiency is due in part to the average pooling operations used between convolutional layers that downsample inputs by averaging over the learned features in patches. Pooling also helps improve accuracy, by reducing the effect of sequencing errors. Finally, pooling helps AMAISE ignore small $k$-mers where host and microbial data might appear more similar; therefore, AMAISE is especially accurate at more difficult tasks (tasks in which the host and microbial sequences look similar). When classifying short read sequences, we do not expect to see the same gains, because of the shorter length and the smaller number of sequencing errors. Thus from both a speed and accuracy perspective, AMAISE is best suited for classifying long-reads. Long-read sequencing technologies are increasingly common[22], and are currently the only option for real-time sequencing applications.

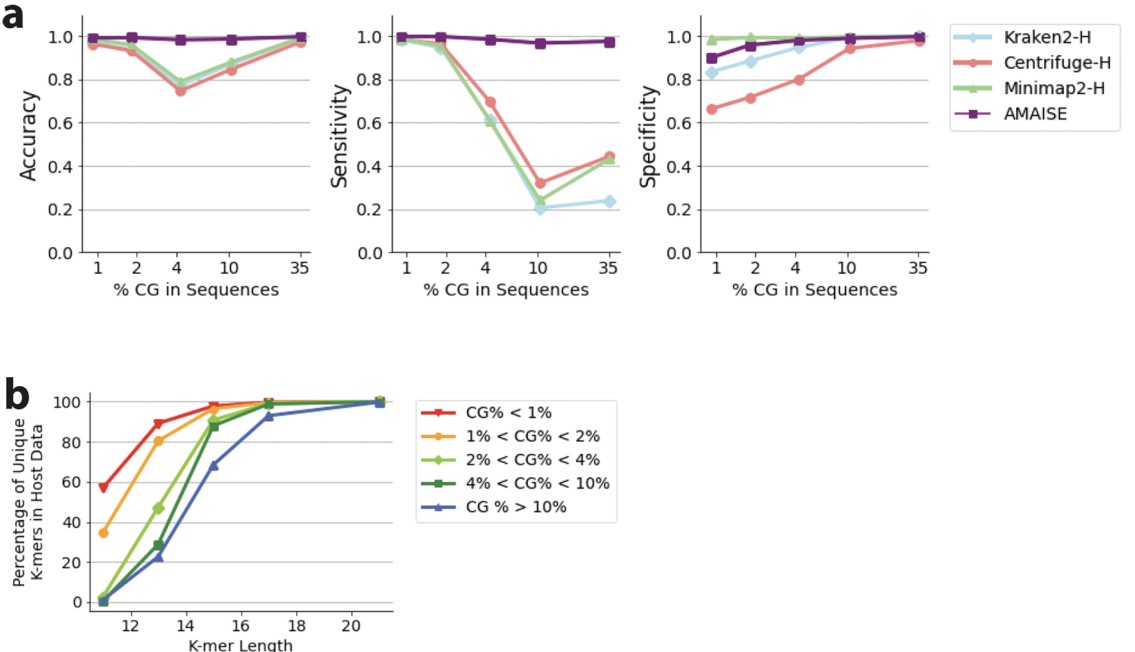

**Fig. 7 Performance of AMAISE, Kraken2-H, Centrifuge-H, and Minimap2-H across samples that varied in terms of 'CG' percentage within the sequences in the sample. a** The performance of AMAISE remains stable across 'CG' percentages (<1%, between 1% and 2%, between 2% and 4%, between 4% and 10%, and above 10%). Kraken2, Centrifuge, and Minimap2's sensitivity drops below 50% when the 'CG' percentage is above 4%. **b** As the percentage of 'CG' in sequences increases, the number of unique 11-mers, 13-mers, and 15-mers in the host data decreases. This suggests that classifying data with lower percentages of 'CG' is an easier task than classifying data with higher percentages of 'CG'.

Thus, the advantage and applicability of AMAISE will likely grow in coming years.

Instead of relying on a reference database at inference time, AMAISE uses a CNN to identify the discriminative patterns. Though CNNs have been used for metagenomic analysis in the past, AMAISE differs from past work in two ways. Past work in classifying sequences either pad all sequences to a long sequence length[14] or truncate all sequences to a short sequence length[13,15]. However, such padding increases classification time, while truncation ignores important information leading to worse accuracy. Second, AMAISE is trained on a combination of reference genomes and Nanopore data. This leads to improved classification performance over training on reference genomes alone[14]; or sequenced data alone[13,15].

Though AMAISE is based on a CNN that is typically referred to as a 'black-box', we leveraged explanation techniques to gain a better understanding of what the model was using to make its predictions. Reassuringly, we found that the model had learned an association between 'CG' and a microbial classification. 'CG' is the least abundant dimer in vertebrate sequences and is more abundant in microbial sequences[23]. This phenomenon is known as "CG suppression". Furthermore, several single base variations from 'CG' were associated with a host classification, aligned with previous work in which such variations have been over-represented in genomes that exhibit 'CG' suppression[23]. It is encouraging that AMAISE captures known biological phenomena (without us explicitly encouraging biological plausibility) and indicates that AMAISE could generalize to species it is not exposed to during training.

However, AMAISE is not solely making its classification decisions based on the 'CG' content in sequences. Our results indicate that not only does AMAISE rely on signals other than 'CG' content, but it relies less on 'CG' content than existing approaches. We found that AMAISE's sensitivity and specificity were more stable and accurate than that of Kraken2, Centrifuge,

and Minimap2 as 'CG' percentage in the test data increased (which corresponded to number of unique $k$-mers in the host test data decreasing). Again, AMAISE's strength compared to other methods is its ability to accurately classify test sets even where the host and microbial data have similar $k$-mers.

AMAISE is not without limitations. First, unlike recent work on Oxford Nanopore's adaptive sequencing, exemplified by software like Readfish[24] and Uncalled[25], which act to classify short base-called sections[24] or raw signals that are not base-called[25] for the purpose of controlling the behavior of the sequencer, AMAISE requires long inputs to be base-called. However, this requirement increases AMAISE's flexibility in that AMAISE's architecture and training scheme could be used to classify sequences from other long-read sequencing technologies that are not capable of adaptive sequencing, like Pacific Biosciences. Second, AMAISE needs VRAM to speed up computation and uses more RAM (has a larger memory footprint) than Centrifuge. However, AMAISE uses less RAM than existing metagenomic classification methods and very little storage. This allows AMAISE to reduce the RAM and storage requirements of those methods by removing their need to store and reference host genomes. This makes AMAISE a useful tool for a user that has access to VRAM but a limited amount of RAM and storage. Furthermore, the amount of RAM and VRAM that AMAISE uses is a parameter that the user can change if they want AMAISE to be more memory efficient. The default parameter makes AMAISE's memory footprint comparable to that of Kraken2 and less than that of Minimap2.

Despite these limitations, AMAISE's ability to classify sequences from different sequencing technologies makes it a versatile tool with wide-ranging applications and a solid base on which to build. Beyond its use in metagenomic classification pipelines, AMAISE can be used as a stand-alone host-depletion tool to remove host DNA in a sample to protect the privacy of the host[26]. Removing human DNA at the source could eliminate

unnecessary exposure to sensitive host genomic data. Given the consistent performance of the proposed approach across a variety of test sets, we expect that AMAISE's default decision threshold will work well for the vast majority of use cases. However, if desired, a user could decrease AMAISE's decision threshold to increase the amount of host DNA that AMAISE removes. AMAISE brings us closer to being able to perform accurate and efficient real-time pathogen detection.

## Methods

**Classification problem setup**. AMAISE learns a mapping $f: \mathcal{X} \rightarrow \mathcal{Y}$, from a feature space $\mathcal{X}$ representing single-end DNA read to a binary valued label space $\mathcal{Y}$ representing the classification of that read (0 for microbe and 1 for host).

AMAISE takes as input a fasta/fastq file of single-end reads and outputs two files: (i) a text file with three columns, where the first column is the ID of each read, the second column contains the classification label of each read, and the third column is the length of the input sequence; and (ii) one fasta/fastq file (the file type depends on the file type of the input) of all the reads that were classified as microbial.

**Datasets**. We trained and validated AMAISE on sequences sampled on random loci (without repetition) from reference genomes and sequences from Nanopore sequencing technology. We tested AMAISE on sequences from Nanopore sequencing technology. The Nanopore sequences used to train and validate AMAISE were from different used for testing. No sequence that appeared in our training set appeared in our validation set and vice versa. However, the species in the training and test sets overlap. The test sets were created first to ensure maximum species variation.

Training data for AMAISE were composed of 242,326 sequences of length 10,000. 50% of those sequences were sampled from reference genomes and the other 50% from Nanopore sequencing technology. 50% of the reference genome sequences were from 3 hosts and the other 50% were equally split among 2951 bacterial, 194 archaeal, 57 fungal, 34 protozoan, and 5455 viral species. 50% of the Nanopore reads were from a human and the other 50% were equally split among 80 bacterial and 8 fungal species.

The bacteria, archaea, fungi, protozoa, and viruses were labeled as microbes, and the mammals were labeled as hosts. We chose the non-host reference genomes from which to sample from the superkingdom lineages of bacteria, archaea, fungi, protozoa, and virus in the following way. We first examined the frequencies of different types of genomic assemblies available in the NCBI RefSeq genomes database using assembly summary files. Observing that we would receive a representation of all phylogenetic lineages, we selected reference and representative assemblies that included complete and chromosomal level representation from multiple species.

For viral genomes, we also selected ICTV species exemplars because first, viral lineages do not have a similar phylogenetic structure to other superkingdoms necessitating more data for training; second, the representative assemblies of the viral lineage collection were small (48 total); and finally, the representative assemblies had a much smaller number of genomic bases than other superkingdoms' assemblies, requiring the acquisition of a large number of genomes to balance their representation with other superkingdoms' assemblies when creating the training set.

We chose three host genomes for training: the human reference genome (GRCh38.p13), the mouse reference genome (GRCm39), and the pig representative genome (Sscrofa11.1). We chose those three because they include the most common organisms and were closest to humans in their evolutionary distance, for which microbiome research is performed.

We applied these selection criteria using R and downloaded genomes that passed. During database creation, we masked their repeat regions with dustmasker[27] using the default parameters, as recommended by the authors of Kraken2[8] and Centrifuge[9].

The Nanopore reads used to train and test AMAISE were obtained from NCBI's Sequence Read Archive (SRA) and the CEPH1463 (NA12878/GM12878, Ceph/Utah pedigree) human genome reference standard on the Oxford Nanopore MinION[28], available from the European Nucleotide Archive under accession PRJEB23027. The reads were not expected to have any post-sequencing modifications (e.g., modifications introduced by read cleaning and quality control procedures). The ground truth labels of all reads except those from the real metagenomic test set were obtained by validating the label in SRA using a Minimap2[10] alignment and only including reads in our sets whose identification matched the label in SRA. The best hits from BLAST[12] were used as ground truth labels for the real metagenomic test set.

Validation data for AMAISE were selected in the same way as the training data and therefore were composed of 242,326 sequences of length 10,000. This validation dataset was used to determine the optimal hyperparameters for AMAISE, the optimal time to stop training AMAISE, and the optimal thresholds to determine AMAISE's classification decisions.

Our 5 test sets with varying host fractions were each composed of 1,000,000 Nanopore sequences with human host percentages varying across 1%, 25%, 50%, 75%, and 99%. The remaining sequences in each test contained bacteria and fungi in the microbial fraction. The bacterial and fungal sequences were equally distributed among 216 bacterial species and 11 fungal species. The length of all the sequences ranged between 82 and 803,815 bp and the median length was 4108 bp.

Our real metagenomic test set contained 800,000 sequences from a set of human oral metagenomic samples publicly available at the NCBI SRA (SRR11547004, SRR11547007). The length of all sequences ranged from 475 to 128,756 bp with the median sequence length being 930 bp.

Our two test sets with non-human hosts either contained a pig or mouse host. These test sets contained 1,000,000 Nanopore sequences each with bacteria and fungi in the microbial fraction. The bacterial and fungal sequences were equally distributed among 216 bacterial species and 11 fungal species. The length of the sequences in the pig test set ranged between 82 and 153,007 bp and the median length was 19,995 bp. The test set contained 99% host data and 1% microbial data. The length of the sequences in the mouse test set ranged between 82 and 204,115 bp and the median length was 4688 bp. The test set contained 99% host data and 1% microbial data.

Our test set with viruses contained 1,000,000 Nanopore sequences with a human host and bacteria, fungi, and viruses in the microbial fraction. The bacterial, fungal, and viral sequences were equally distributed among 216 bacterial species, 11 fungal species, and 52 viral species. The length of all the sequences ranged between 82 and 803,815 bp and the median length was 4102 bp. The test set contained 99% host data and 1% microbial data.

To obtain our test sets with varying amounts of 'CG', we stratified our synthetic human host test set with 50% host, 50% microbial data into equal-sized quintiles by 'CG' percentage. The sequences were binned based on whether they had <1% 'CG', between 1% and 2% 'CG', between 2% and 4% 'CG', between 4% and 10% 'CG', and <10% 'CG.'

The final training and validation sets were each 75 GB. All test sets were 16 GB except the real metagenomic test set, which was 3 GB. To facilitate reproducibility, the accession codes needed to download the reference genomes and Nanopore sequences used to create the training, validation, and test sets are available on Github (https://github.com/MLD3/AMAISE).

**Data preprocessing**. To create the training sets, sequences of length 10,000 were randomly sampled from the files containing reference genomes and Nanopore sequences. Each sequence was converted into a $L \times 4$ matrix where $L$ is the length of the sequence and $A = [1, 0, 0, 0]$, $C = [0, 1, 0, 0]$, $G = [0, 0, 1, 0]$, and $T = [0, 0, 0, 1]$.

The training data were stored in a $N \times L \times 4$ matrix where $N = 242,326$ and $L = 10,000$. The validation data were stored in a $N \times L \times 4$ matrix where $N = 242,326$ and $L = 10,000$. These matrices were used to train and validate AMAISE.

At inference time, for efficiency, sequences were binned in increments of 50, and sequences within the same bin were saved to a text file and truncated to the smallest sequence in that bin. For sequences above length 5000, the bin size was 1000. All sequences that were greater in length than 6000 were truncated.

AMAISE wrote the sequences in the same bin into text files. Once AMAISE wrote the maximum amount it could to memory, we iterated through the text files and encoded each input read into a numerical matrix via one-hot encoding. More specifically, we converted $N$ DNA sequences of length $L$ into a $N \times L \times 4$ matrix, where $A = [1, 0, 0, 0]$, $C = [0, 1, 0, 0]$, $G = [0, 0, 1, 0]$ and $T = [0, 0, 0, 1]$.

**Model architecture**. AMAISE is based on a CNN that takes as input a sequence and outputs a classification label for that sequence. When classifying sequences, recurrent neural networks are commonly applied. In contrast, we used a CNN since they do not rely on a sequential relationship between input data and can therefore forward propagate (i.e., perform inference) faster[7]. Furthermore, we saw a similarity between convolution layers and $k$-mer-based methods like Kraken2. K-mer0based methods classify reads by matching $k$-mers from the reads to $k$-mers in their databases. Convolutional layers learn the parts of DNA reads that are most important for classification and store them in filters.

We selected AMAISE's model architecture and training parameters (specifically the number of filters, the size of the filters, the number of convolutional layers, the number of fully connected layers, the size of the average pooling in between convolutional layers, the dropout between convolutional layers, the learning rate, and the weight decay for L2 regularization) based on the validation loss using random grid search with a budget of 20 (see Supplementary Note 6, Supplementary Table 1 for a full list of hyperparameters and ranges over which we searched).

After tuning, AMAISE's final model architecture consists of four convolutional layers, a global average pooling layer, and one fully connected layer. By including a global average pooling layer AMAISE applies to reads of varying lengths without requiring any additional preprocessing (e.g., padding). Each convolutional layer in AMAISE contains 128 filters of length 15. We applied a rectified-linear unit activation function and an average pooling layer of length 5 after each convolutional layer. We used two forms of regularization when training AMAISE: L2 regularization with $\lambda = 1e-5$ and gradient norm regularization. We used gradient norm regularization to ensure the accuracy of our interpretability method[29].

```
Inputs: InputFile, input file of sequences
Outputs: ml_probs.txt: text file with classification output of AMAISE per each sequence
in InputFile; mlpaths.fasta/fastq: a fasta/fastq file (the file type depends on the file type
of the input) of all the reads that were classified as microbial
Predefined Variables: FileLengths = List of lists, where each list contains the lengths
of sequences to store in text files; BatchSize = size of batch of sequences to classify at
once; AMAISE: CNN to classify sequences

 1: for each list in FileLengths do
 2:     for each length in sorted(list) do
 3:         for each sequence in InputFile do
 4:             if length(sequence) < seq_cutoff and length(sequence) - inc1 ≤ length
    then
 5:                 store sequence in length.txt
 6:             else
 7:                 if length(sequence) ≥ seq_cutoff and length(sequence) - inc2 ≤ length
    then
 8:                     store sequence in length.txt
 9:                 end if
10:             end if
11:         end for
12:     end for
13:     for each TextFile in all stored text files do
14:         CurrentBatchSize = 0
15:         Sequences = []
16:         for each sequence in Textfile do
17:             if CurrentBatchSize != BatchSize then
18:                 Store sequence in Sequences
19:                 CurrentBatchSize += 1
20:             else
21:                 Outputs = AMAISE(Sequences)
22:                 Store Outputs in ml_probs.txt
23:                 for i in length(Outputs) do
24:                     if Outputs[i] < Threshold then
25:                         Write InputFile[i] to ml_paths.fasta/fastq
26:                     end if
27:                 end for
28:                 CurrentBatchSize = 0
29:                 delete TextFile
30:             end if
31:         end for
32:     end for
33: end for
```

**Fig. 8 Pseudocode describing Input preprocessing.** To classify sequences in an input file, we first sort the sequences efficiently by binning the sequences by length and storing sequences of similar length in a single text file. We then classify the sequences in each text file using AMAISE. We input the sequences to AMAISE in batches and delete a text file after classifying all the sequences in it.

Initially, AMAISE outputs a probability that each input read is from a host. We then threshold each probability to map it to a binary classification label. Optimizing for sensitivity and specificity, we selected thresholds based on the validation data. We calculated a different probability threshold for sequence lengths 100, 150, 200, 300, 500, 1000, 5000, and 10,000. These thresholds are in Supplementary Note 6, Supplementary Table 2. For each input sequence, we selected the threshold that corresponded to the closest length.

**Training details**. To learn the model parameters, we optimized the cross-entropy loss on the training data, using Adam[30]. We concatenated the training sequences to lengths 100, 1000, and 10,000 so that AMAISE could be exposed to sequences of multiple lengths. We initialized training using a learning rate of $1e-3$ with no decay, training until the loss on the validation data did not improve after 30 epochs. All model training was implemented using Pytorch on a server with 64 AMD EPYC 7702 64-Core Processors (128 hyperthreaded cores), 256 GB of RAM, and 8 NVIDIA RTX 2080 GPUs.

**Baseline machine learning classification tools**. We compared AMAISE to the following baseline CNN-based approaches from the literature: a CNN that pads all inputs to the maximum length[14] and a CNN that truncates all inputs to the minimum length[13]. We compared AMAISE to the following baseline CNNs to test the benefit of training on sequences sampled from reference genomes and Nanopore data: a CNN that trains solely on sequences from reference genomes[14] and a CNN that trains solely on sequences from Nanopore data[13,15].

**Software versions and databases of baseline metagenomic classification tools**. On the test sets, we compared AMAISE to Kraken2[8], Centrifuge[9], and Minimap2[10]. For Kraken2-H, Centrifuge-H, and Minimap2-H, we created indices with the human reference genome. If a read in the test set did not match or was not aligned to that index, then we did not consider that read to be from a host. For Kraken2-M and Centrifuge-M, we created databases with the 2951 bacterial reference genomes, 194 archaeal reference genomes, 57 fungal reference genomes, 34 protist reference genomes, and 5455 viral genomes that we used to train

AMAISE. For Kraken2-HM and Centrifuge-HM, databases also included the human reference genome. We briefly describe each baseline below:

*Kraken2 (Version 2.1.0):* Given an input sequence, Kraken2 maps the $k$-mers (genomic substrings of length $k$) in that sequence to individual genomes. It then aggregates the assignments to the lowest common ancestor taxonomy to make a classification label. It was designed to quickly and accurately classify DNA sequences while being more memory efficient than its predecessor Kraken. Despite Kraken2 being more memory efficient than Kraken, it is not optimized in the same way as Centrifuge, so we did not expect Kraken2 to be able to run in our resource-constrained environment[8].

*Centrifuge (Version 1.0.4):* The classification strategy for Centrifuge is similar to that of Kraken2. However, Centrifuge uses a different data structure than Kraken2 to store the $k$-mers of individual genomes and classify input sequences. This data structure has a relatively small memory footprint that is based on the Burrows–Wheeler transform and the Ferragina–Manzini index. It was designed to achieve comparable speed and accuracy to existing methods while being more memory efficient. Since Centrifuge was designed to be memory efficient, we expected Centrifuge to run in both computational environments (described below)[9].

*Minimap2 (Version 2.17):* Minimap2 is a fast and accurate alignment program. Alignment programs are designed to align reads to a closely related, single reference genome. They are unable to align reads to a large group of genomes and cannot perform multi-class classification. As a result, we only use Minimap2 as a baseline for our host depletion comparison by aligning inputs to the human (host) genome[10]. To use Minimap2 as a host depletion tool, we converted its best alignment to a classification label in the following way. If Minimap2 could align an input sequence to the host genome, we considered Minimap2's classification of that sequence to be "host." Else, we considered Minimap2's classification of that sequence to be "microbe." Minimap2 was designed to align both short and long reads, but its strength compared to other aligners is its ability to align long, error-prone reads to a closely related reference genome accurately and quickly so that the genomic variations in the reads can be quantified. Minimap2 achieved comparable accuracy to existing short-read aligners and higher accuracy than long-read aligners. We did not expect Minimap2 to be able to run in our resource-constrained environment since it is not optimized to be memory efficient[10].

**Testing computational environments**. We evaluated our proposed approach to the baselines in two different computational environments. The first was the same environment in which the model was trained. However, we limited ourselves to using only one out of the 8 NVIDIA RTX 2808 GPUs for evaluation. The second was a GCP VM Instance running an N1 Machine with Debian GNU/Linux 10 (buster) with 4 vCPUs, 1 Tesla T4 GPU, 16 GB of RAM, and 200 GB on the boot disk. The T4 GPU has 320 NVIDIA Turing Tensor Cores and 2560 NVIDIA CUDA cores. The N1 Machines can be created with four different types of GPUs: the NVIDIA Tesla K80 GPU, the NVIDIA Tesla P4 GPU, the NVIDIA Tesla T4 GPU, and the NVIDIA Tesla V100 GPU. The NVIDIA Tesla T4 GPU's performance was comparable to several laptop GPUs, including the NVIDIA Geforce GTX 1660 Laptop GPU[31], so we created our Virtual Machine Instance with the T4 GPU. We ran each tool such that they used 16 threads for computation.

**Evaluation metrics**. Our goal was to create a diagnostic tool that was accurate, cheap, and fast. When comparing AMAISE to existing in silico host depletion tools, we evaluated methods' overall accuracy using classification accuracy, sensitivity, and specificity. When calculating classification accuracy, sensitivity, and specificity, we assigned hosts with the positive label and microbes with the negative label. When comparing AMAISE + Centrifuge-M and AMAISE + Kraken2-M to Centrifuge-HM and Kraken2-HM, we evaluated methods' overall accuracy using host accuracy and multi-class microbial accuracy. The accuracy of each pipeline on the sequences labeled as host was computed by the percentage of sequences that were correctly labeled as "Homo sapiens" out of the number of sequences with the true label "Homo sapiens." The accuracy on the sequences labeled as microbial was computed by the weighted average of the number of sequences that were labeled as a certain microbe whose true label was that microbe.

For both our host depletion and metagenomic classification pipeline comparisons, we used maximum resident set size (which is a good approximation for the amount of RAM for a process), and storage requirements to evaluate each methods' cost; and elapsed wall clock time to evaluate each methods' overall classification time. We modeled our classification time and part of our maximum resident set size evaluation on the Kraken2's evaluation[8]. We ran each host depletion method using 16 threads on the same number of reads. We used the taskset command to restrict the number of processors each method was allowed to use. We used the "/usr/bin/time -v" command to get the elapsed wall clock time and the maximum resident set size. AMAISE also used VRAM via the GPUs, so we added the MiB of VRAM that AMAISE used to the amount of RAM that it used. We calculated the amount of VRAM that AMAISE used with the command "nvidia-smi." We evaluated storage requirements using the "ls -l" command in the folders containing the existing host depletion methods' databases and indices and the folder containing AMAISE. We did not include the storage necessary for the packages needed to run the baseline methods and AMAISE.

**Interpretation method overview**. We applied the DeepLift algorithm from Captum, Pytorch's library for Model Interpretability, to a Nanopore read set to identify the $k$-mers that contributed to AMAISE's classification decision[19]. Briefly, DeepLift determines the contribution of each element of an input sequence to the final classification by approximating the gradient of the model output with respect to each element of the input in the following way. It compares the activation of each neuron in the model given the current input to its activation given a reference input. Based on the difference in values, the DeepLift algorithm computes a contribution score for each neuron and uses those contribution scores to compute the final contribution for each element of the input.

We applied DeepLift to a randomly selected subset of 2356 Nanopore sequences, 50% of which were from a human and 50% of were equally distributed among 216 bacteria and 11 fungi. We used 20 different references, where each reference was a permuted version of the input sequence, and we averaged the output contributions to get the final contribution of each nucleotide in the input sequence[32].

Once we calculated the input attributions for a sequence, we summed the attributions from each nucleotide in a $k$-mer (where $k = 15$, the size of the filters in AMAISE's first convolutional layer) to calculate the amount that each $k$-mer in the sequence contributed to the final classification decision. We sorted the $k$-mers by the mean attribution and selected the 5 $k$-mers with the highest and lowest mean attribution for further analysis.

**Pseudocode of input preprocessing**. We preprocess input sequences before feeding them into AMAISE. The preprocessing that we perform is described in pseudocode in Fig. 8.

**Description of code functionality**

*System requiremenrts*. AMAISE requires a computational environment with a Linux-based machine/Ubuntu that has a GPU and enough RAM to support downloading the necessary packages and GPU driver and to support the in-memory operations.

AMAISE also requires pip and Python 3.6 or above. Required packages and versions are listed in "requirements.txt". If one's environment is already set up to run code on a GPU, one can install the packages in requirements.txt using: pip install -r requirements.txt If one's environment is not set up to run code on a GPU, in the folder "helpful_files", we have included a bash script "startup.sh" that contains the code that we used to install GPU drivers onto a Google Cloud Platform Virtual Machine that may help install GPU drivers. "startup.sh" will have to be altered to download the correct drivers for one's machine and to set up the appropriate paths to install the packages.

*Usage notes for AMAISE*. AMAISE bins input data by length to efficiently and accurately process sequences in batches. To alter the constants used to perform this binning, one can alter the following variables in "constants.py":

- seq_cutoff: the sequence length at which to change the increments in which sequences are binned from inc1 to inc2
- inc1: the increment in which sequences whose length is less than seq_cutoff are binned
- inc2: the increment in which sequences whose length is greater than seq_cutoff are binned
- max_len: the maximum length that all sequences will be truncated to

Their default values are seq_cutoff = 5000, inc1 = 50, inc2 = 1000 and max_len = 6000.

The total sequence length that AMAISE can write to memory, lim, is a parameter that the user can set in "constants.py." Its default value is lim = 1, 200, 000, 000. If lim is less than the total length of the input, we keep track of the maximum sequence length that we were able to write to memory before we ran out of memory, classify the sequences that we have written to text files, and then iterate through the input fasta/fastq file and store the sequences that are larger than the ones we have already classified until we run out of memory again.

The batch size $N$ that we used for testing is batch_max_len/$L$, where batch_max_len is a constant that the user can set in "constants.py." Its default value is batch_max_len = 900, 000. Increasing or decreasing $N$ will increase or decrease the amount of RAM and VRAM that AMAISE uses.

To alter the thresholds that AMAISE uses for classification, one can alter the variable threshs in "helper.py".

Demo test data is in the folder "demo_test_data."

The outputs that one should expect from running "host_depletion.py" is in the folder "correct_outputs." One can check the output files beginning with "mlprob" and "mlpath" against the outputs in "correct_outputs" by inputting the output file one generated and the corresponding output file in "correct_outputs" into https://www.diffchecker.com/.

To use AMAISE, use the command python3 host_depletion.py -i < inputfile > -t <typefile > -o < outfolder> To classify the demo test data, run python3 host_depletion.py -i demo_test_data/nanopore_demo_data.fastq -t fastq -o single_end_output where the input arguments are:

- i: the reads in a fasta or fastq file
- t: the type of file that the reads are in (fasta, fastq)

- o: the name of the folder that one wants to put the outputs in and the outputs (in <outfolder>) are:
- mlprobs.txt: a text file with three columns, where the first column is the ID of each read, the second column contains the classification label of each read, and the third column is the length of the input sequence
- mlpaths.fasta or mlpaths.fastq: a fasta/fastq file (the file type depends on the file type of the input) of all the reads that were classified as microbial

*Reproducing the analyses in the text*. The commands one needs to be able to run to reproduce the analyses in the text are "time -v", "taskset", "ls -l", and "nvidia-smi."

To reproduce the analyses in the text on the demo data, one should first run "single_end_AMAISE_evaluation.py" after changing "track_gpu" in "constants.py" to True. This will calculate the evaluation metrics including the amount of VRAM that AMAISE uses and store the calculations. Tracking GPU usage does slow down the code, so to calculate the speed that AMAISE runs in without tracking GPU usage, re-run "single_end_AMAISE_evaluation.py" after setting "track_gpu" in "constants.py" to False. The evaluation metrics are written to "<outfolder>/single_end_resource.txt."

The outputs that one should expect from running "single_end_AMAISE_evaluation.py" is in the folder "correct_outputs." The speed and peak memory usage that one gets may differ from the speed and peak memory usage reported in "correct_outputs" since this is dependent on the computational environment that one runs the code in. The results in "correct_outputs" are from running the code on a server with 64 AMD EPYC 7702 64-Core Processors (128 hyperthreaded cores), 256 GB of RAM, and 8 NVIDIA RTX 2080 GPUs.

*Run time on "Normal" desktop computer*. We mimic a "Normal" desktop computer with a Google Cloud Platform Virtual Machine that has 4 vCPUs, 16 GB of RAM, 200 GB of boot disk storage, and 1 Tesla T4 GPU with 320 NVIDIA Turing Tensor Cores and 2560 NVIDIA CUDA cores[18].

*Run time for classifying test sets*: AMAISE takes 44–45 min to classify Nanopore test sets with 1 million sequences.

*Run time for classifying demo data*: AMAISE takes 42 s to classify the demo Nanopore data.

**Reporting summary**. Further information on research design is available in the Nature Research Reporting Summary linked to this article.

## Data availability

The reference genomes used to train AMAISE are available on NCBI. Their accession codes are on GitHub (https://github.com/MLD3/AMAISE). The Nanopore test sets that we used to train and evaluate AMAISE are downloaded from NCBI's Sequence Read Archive and the CEPH1463 (NA12878/GM12878, Ceph/Utah pedigree) human genome reference standard on the Oxford Nanopore MinION created by Jain et al., which is available from the European Nucleotide Archive under accession PRJEB23027. The source data underlying the plots presented in the main text is available in Supplementary Data 1.

## Code availability

The source code for AMAISE is available on Github (https://github.com/MLD3/AMAISE).

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

## Acknowledgements

This work was supported by the D. Dan & Betty Kahn Foundation through a grant to U.-M. M.K. received funding from the Graduate Fellowship for STEM Diversity. R.P.D. received funding from the National Heart, Lung, and Blood Institute (R01HL144599). J.R.E. received funding from the National Institute of Allergy and Infectious Diseases (R01AI129958) and the National Heart, Lung, and Blood Institute (R01HL144599). The views and conclusions in this document are those of the author and should not be interpreted as necessarily representing the official policies, either expressed or implied, of any funding sources.

## Author contributions

M.K. and J.W. conceived the study. P.R. acquired the data for training and testing AMAISE and baseline methods; and created the reference indices for Kraken2 and Centrifuge. M.K. created the training and test data sets for AMAISE and the baseline methods; designed the architecture and training scheme of AMAISE; and implemented AMAISE and the baseline methods on a server and Google Cloud Platform Virtual Machine. M.K., P.R., J.R.E., R.P.D., and J.W. analyzed and interpreted results. M.K. drafted and prepared the original and revised manuscript. M.K., P.R., J.R.E., R.P.D., and J.W. discussed the results and contributed to the original and revised manuscript.

## Competing interests

The authors declare the following competing interests: R.P.D. is the cofounder of Sequal Inc. The remaining authors declare no competing interests.
