## [Peer Review File · Communications Biology]

Reviewers' comments:

Reviewer #1 (Remarks to the Author):

Dear Authors

AMAISE: A Machine Learning Approach to Index-Free Sequence Enrichment

The manuscript details a machine learning approach, based on a convolutional neural network, to detect contamination/host sequences in metagenomic samples. AMAISE aims to circumvent costly and time consuming laboratory methods and is proposed to be an alternative to currently used reference dependent in silico methods.

The application of machine learning methods in biological sciences, molecular biology and in particular metagenomics is of critical importance. These methods will greatly assist in harnessing the inherent power and predictive capabilities of biological data which includes sequencing data.

Some comments, suggestion and questions regarding AMAISE:

1. The test set is very sparse and consists of only one set for Illumina and one for Nanopore with both being artificially constructed. Furthermore the test set represents 99% human. What about creating test sets for different levels of human contamination, e. g. 95%, 90%, 75% etc?
2. The size of test sets further do not represent current metagenomic data sizes. 1,000,013 paired-end Illumina sequences, 1.2 GB, can hardly be regarded as a true reflection of the amount of data generated per sample in a metagenomic study.
3. No tests or reporting of tests done using "real" metagenomic data is reported. There is a wealth of public metagenomic data available for testing. Why were no comparisons done using some publicly available data?
4. "The two read sets were assembled from sequencing reads obtained from the NCBI SRA, and did not have any post-sequencing modifications (e.g., modifications introduced by read cleaning and quality control procedures).", "However, recall that the reads in our test set did not undergo quality control." - In general, researchers will do sequence quality control as the first part of any project/analysis. The test set as such does not reflect true research protocols or pipelines. This further touches on the point above where the test sets are artificial and no test were done on "real" metagenomic data sets. In the manuscript the poor accuracy of Bowtie2 using raw sequencing data is highlighted and the accuracy improved after quality filtering. It seems that due to the inclusion of Nanopore data you were hesitant to include an initial quality filtering step in the testing procedure.
5. "Though CNNs have been used for metagenomic analysis in the past [7, 21-26]" - Why were no tests done against other CNN host/contamination depletion tools/applications currently available? If there are other CNN host/contamination depletion tools/applications currently available, what makes AMAISE novel?
6. "First, it is designed to efficiently and accurately process sequences of variable length." - I assume that the variable length portion refers to Nanopore data. Is AMAISE the only program available to handle Nanopore data? Is this where the novelty of AMAISE lies?
7. "Second, AMAISE is trained on a combination of reference genomes, Nanopore and Illumina data." - Again the highlight on Nanopore data.
8. If the novelty and purpose of AMAISE is Nanopore data, I would suggest repurposing and redirecting the manuscript as a Nanopore/ long read applications.
9. In point 7 and in the text training is implied on a combination of reference genomes. In the manuscript it is stated that "We chose those three because they include the most common organisms and were closest to humans in their evolutionary distance, for which microbiome research is performed". Human, mouse and pig genomes were used for the training but only human was included in the validation and testing. Why not the other 2 which are "the most common"?
- 10 "1,000,013 paired-end Illumina sequences, 99% of which are from a human and 1% of which are equally distributed among 16 bacteria and 3 fungi. 1,017,000 Nanopore sequences, 99% of which are from a human and 1% are equally distributed among 216 bacteria and 11 fungi." - Both test sets did not include viruses. Why not?

11. "This represents a 300-fold improvement in storage efficiency over common classifications or alignment baselines." – Again, not compared to other CNN applications. AMAISE is never explored or compared against other metagenomic CNN algorithms or machine learning approaches currently available.

12. "AMAISE can be used in a number of different ways. Here, we investigate two specific applications i) as a stand-alone host-depletion tools and ii) in combination with existing metagenomic classifiers." – No explanation of the number of different ways. The 2 ways as in the text are the same thing. AMAISE does host depletion and not classification. Point 2 is just an extension of the first part where classification is done on a host depleted set. Obviously, the other programs will perform better if an initial host depletion was done already.

13. "Reassuringly, we found that the model had learned an association between CG and a microbial classification." - In essence, this model discriminates host and non-host based on GC content of a sequence. The question arises if classifying a sequence as host or non-host based on GC is novel enough.

14. "AMAISE is not without limitations. First, AMAISE had lower specificity than Kraken2-H, Centrifuge-H, and Bowtie2-H on Illumina reads." – Only effective on nanopore reads. The low specificity may be due to sequence length and GC based on sequence length. The short Illumina sequences do not provide a strong enough signal to detect a true negative.

15. "Finally, AMAISE was slower than Kraken2-H on both test sets, and Centrifuge-H on Illumina reads." – Not really that effective.

16. "However, a user can tune AMAISE's decision threshold to increase specificity to microbial reads if privacy protection is not necessary", "Depending on the risk tolerance, a user can tune AMAISE's decision threshold to increase its sensitivity to host DNA." - In both these sentences there a 2 mentions which are worrisome. i) It seems like the user will have to tweak/tune the application when working with Illumina data. Can does not be done by the authors and incorporated in the model? Are these parameters/tuning efforts based on the composition of a sample and will it differ across samples? ii) The mention of "privacy protection" and "Depending on the risk tolerance". This needs to clarified and is currently very ambiguous.

17. The bar plots are of poor quality and can surely be improved.

In summary:

It would appear that AMAISE is more suited for Nanopore data as the longer reads strengthens the GC signal used by the model to classify. Testing of the model should definitely be expanded and the novelty of AMAISE be articulated. The authors need to appreciate the fact that not all users of AMAISE will have a bioinformatic/computational background or similar experience and that expecting users to tweak/tune settings for Illumina data will not always be possible without the appropriate guidelines. Is it not possible to iron out these "tuning" parameters and incorporate. It would be troublesome if reproducible results are not obtained and if parameters need to be adjusted according to expected sample composition when using Illumina data. The authors need to clearly explain phrases such as "Depending on the risk tolerance" and "privacy protection is not necessary".

All the best

Reviewer #2 (Remarks to the Author):

This article is about a new method to separate host-microbiota reads in a metagenomic set. I found this idea profoundly interesting since I have faced this problem by analyzing microbiomes mixed with unknown host reads. This is an important issue, microbial ecology has followed a significant improvement with NGS technologies, but all this host-genome DNA drags the study of non-human host-microbial systems metagenomic sets.

However, the authors focus on the importance of their technique for clinical data; thus, the host = human. The authors claim AMAISE is more effective than other kraken2, minimap2, and bowtie2.

AMAISE advantage is it is reference-free and fast. Nevertheless, the host reference genome known I do not see the point of adding a new bioinformatic tool, kraken2, can do the job.

Reviewer #3 (Remarks to the Author):

The authors describe AMAISE, a method and software tool that uses a convolutional neural network (CNN) to classify metagenomics reads according to whether they originated from a host organism. The CNN classifier is trained on examples drawn from reference genomes and from many sequencing reads. The experiments show that AMAISE achieves superior classification accuracy for Nanopore sequences and comparable accuracy for Illumina sequences compared to other approaches. The experiments also show that AMAISE has a similar memory footprint to Kraken2, but is slower.

The idea is novel and interesting and the manuscript is very well written.

Major comments

The AMAISE CNN is trained on various sequences drawn from a combination of reference genomes and reads. This gives AMAISE, as deployed here, visibility into genetic variation and sequencing error. This is in contrast to the k-mer-based approaches which, as deployed here, know only about reference genomes. On the one hand, we can consider this an inherent advantage of AMAISE. But on the other, there is nothing keeping the authors from also including genetic variation in the competing k-mer-based indexes. E.g. the authors could use more than one human reference genome in the k-mer indexes. Without such a comparison, it is hard to tell whether we're seeing an inherent advantage of AMAISE or an advantage of giving the classifier a "peek" at genetic variation. Note that some published index-based metagenomics classifiers are specifically geared toward using pan-genome indexes, and have been applied in host sequence removal scenario using multiple human genomes [doi:10.1016/j.isci.2021.102696].

The authors explain the lower accuracy of Bowtie 2 by stating "the reads were not subjected to quality control, leaving adapter contamination or poor quality bases." Rather than use this tentative explanation, the authors should run Bowtie 2 with the --local option, which causes it to soft-clip low-quality or other poorly-matching bases from the extremes of the read.

A stated advantage of AMAISE is that it doesn't require a "memory-intensive" index data structure. But we read later that "In terms of peak memory usage, on both datasets, AMAISE was on par with Kraken2-H and Bowtie2-H, requiring 3.0 GB of RAM and 1.6 GB of video RAM (VRAM)." Not only does AMAISE's memory footprint seem comparable to methods that use a "memory-intensive" index (indeed, AMAISE's footprint is much larger than Centrifuge's), it also requires a specialized type of RAM (VRAM) that may not be as readily available to users. Is this really a good deal? The claims and discussion in the manuscript should be much more detailed and nuanced around this point.

Minor comments

Given that the explanation about what the model is learning revolves around the presence of CpGs, should we be concerned that the model will be differentially effective at classifying host (e.g. human) sequence from CpG islands? This would seem to be another relevant point for contrasting the CNN with index-based methods.

Related to the previous point: do the authors expect that the AMAISE method will also work well in scenarios where the host genome and the target genome have a similar overall level of CG content?

Why is AMAISE's memory footprint so large? Do the CNN parameters take gigabytes of RAM?

The tools were evaluated on a test set that used a 99-to-1 mix of host versus non-host reads. Can the authors say anything about whether they expect AMAISE's relative performance to change much when a different mix is used? (E.g. 90-to-10 or 999-to-1?)

Authors' response to reviews

Title: AMAISE: A Machine Learning Approach to Index-Free Sequence Enrichment

Version: 2

Date: March 10, 2022

Author's response to review: see over

Title: AMAISE: A Machine Learning Approach to Index-Free Sequence Enrichment

Version 1 Date: August 24, 2021

Version 2 Date: March 10, 2022

Dear Dr. LaFlamme and the Nature Communications Biology Editorial Board,

Thank you for considering our manuscript (COMMSBIO-21-2272-T) to the journal *Nature Communications Biology* and inviting us to submit a revised version for further consideration.

In response to the critiques of our manuscript received on August 24, 2021, we have significantly revised our manuscript:

1. We have reorganized the content of our manuscript to focus entirely on classifying long-read data (as opposed to short- and long-read data).
2. We have emphasized the novelty of our approach by including additional experiments comparing it to convolutional neural network-based approaches from past literature (pg 5-6, line number 108-123).
3. We constructed and included additional analyses on the performance of our proposed approach and existing methods when applied to
 - a. 5 test sets with human host percentages (in percentage of number of reads) varying across 1%, 25%, 50%, 75%, and 99% (pg 6, line number 126-149; pg 6-7, line number 180-184; pg 8-9, line number 185-229)
 - b. 1 test set with real metagenomic data (pg 6-7, line number 151-170; pg 6-7, line number 180-184)
 - c. 2 test sets with non-human hosts (Supplementary Text pg 1, line number 2-9)
 - d. 1 test set with viruses in the microbial fraction (Supplementary Text pg 2, line number 10-15)
 - e. 5 test sets with differing percentages of 'CG' in the sequences (pg 11-12, line number 279-299)

We note that the Nature Communications Biology Revision Checklist states to “convert all bar graphs to box-and-whisker or dot-plot format to show data distribution.” Some of our results remain plotted as bar graphs because the datasets are so large that error bars are meaningless.

We believe we have thoroughly addressed the Reviewers' comments and have strengthened the manuscript. We appreciate the Reviewers' efforts and believe that by addressing the queries, our manuscript has significantly improved.

We have detailed our revisions and responses to the comments below. Please note that all references to page numbers and line numbers refer to the version with all track changes accepted in “AbstractMainMethods.docx “. In addition to those edits outlined in the response below, minor edits have been made throughout to ensure consistency.

Reviewer 1

Reviewer 1.1. If the novelty and purpose of AMAISE is Nanopore data, I would suggest repurposing and redirecting the manuscript as a Nanopore/ long read applications. "Second, AMAISE is trained on a combination of reference genomes, Nanopore and Illumina data." - Again the highlight on Nanopore data.

Response 1.1. We thank the reviewer for this very useful feedback. We agree that the advantage of AMAISE is most evident when applied to long-read sequencing data in comparison to other tools. Thus, we have reorganized our manuscript to focus entirely on the Nanopore/long read application. Specifically, we:

- focus all experiments on Nanopore data
- revised the Discussion to include a discussion regarding the utility of AMAISE for real-time sequencing and why AMAISE is better suited for long-read sequences

New content in Discussion (pg 13, line number 323-334)

AMAISE is especially efficient and accurate at classifying long-read complex data. Its efficiency is due in part to the average pooling operations used between convolutional layers that downsample inputs by averaging over the learned features in patches. Pooling also helps improve accuracy, by reducing the effect of sequencing errors. Finally, pooling helps AMAISE ignore small k-mers where host and microbial data might appear more similar; therefore, AMAISE is especially accurate at more difficult tasks (tasks in which the host and microbial sequences look similar). When classifying short read sequences we do not expect to see the same gains, because of the shorter length and the smaller number of sequencing errors. Thus from both a speed and accuracy perspective, AMAISE is best suited to classify long-reads. Long-read sequencing technologies are increasingly common [22], and are currently the only option for real-time sequencing applications. Thus, the advantage and applicability of AMAISE will likely grow in coming years.

New content in Discussion (pg 14, line number 364-370)

AMAISE is not without limitations. First, unlike recent work on Oxford Nanopore's adaptive sequencing, exemplified by software like Readfish [24] and Uncalled [25], which act to classify short base-called sections [24] or raw signals that are not base-called [25] for the purpose of controlling the behavior of the sequencer, AMAISE requires long inputs to be base-called. However, this requirement increases AMAISE's flexibility in that AMAISE's architecture and training scheme could be used to classify sequences from other long-read sequencing technologies that are not capable of adaptive sequencing, like Pacific Biosciences.

Reviewer 1.2. The test set is very sparse and consists of only one set for Illumina and one for Nanopore with both being artificially constructed. Furthermore the test set represents 99% human. What about creating test sets for different levels of human contamination, e. g. 95%, 90%, 75% etc?

Response 1.2. We have added 5 additional Nanopore test sets that have the following levels of human composition (1%, 25%, 50%, 75%, 99%). AMAISE's performance remained stable across test sets and was either comparable or better than that of Kraken2, Centrifuge, and Minimap2. We have updated the Results and Methods sections as described below.

New content in Results (pg 5, line number 95-98)

In our samples, we first varied the percentage of reads of the sample that pertained to host. Specifically, we constructed 5 test sets with human host percentages (in percentage of number of reads) varying across 1%, 25%, 50%, 75%, and 99%, and each with bacteria and fungi in the microbial fraction.

New content in Results (pg 6-7, line number 138-184)

Across test sets with different host fractions, AMAISE achieved the highest sensitivity (>99%) compared to Kraken2-H, Centrifuge-H, and Minimap2-H. The existing method with the closest sensitivity, Centrifuge-H, achieved a sensitivity of only 87.5%. In addition, AMAISE achieved comparable or higher specificity and accuracy (> 98%) compared to existing methods. Across test sets with different host fractions, AMAISE ran in less than 20 minutes, on par with Kraken2-H and Minimap2-H, and between 10 and 50 minutes faster than Centrifuge-H (Figure 2).

Figure 2. Across samples that varied in terms of host percentage, the performance of AMAISE remained stable. AMAISE consistently achieved higher accuracy while requiring less storage and remaining competitive with respect to classification time and peak memory usage compared to other methods. Gains in accuracy and speed over Kraken2-H, Centrifuge-H, and Minimap2-H are most significant when the fraction of host samples is above 25%.

...

In terms of peak memory usage, on all datasets, AMAISE was on par with Kraken2-H, requiring 3.0 GB of RAM and 1.6 GB of video RAM (VRAM) compared to Kraken2-H's requirement of 4 GB of RAM. However, AMAISE required significantly less RAM than Minimap2-H. Finally, on all datasets, AMAISE required less than 0.3% of the storage required by the other existing approaches (Figure 2-3).

New content in Results (pg 8-9, line number 204-229)

Across test sets with different host fractions, the pipelines including AMAISE had higher host accuracies and microbial accuracies than the pipelines without AMAISE (Figure 4). AMAISE had the greatest impact on host sequence classification, improving the host accuracy of the pipelines from 85% (Centrifuge-HM) and 84% (Kraken2-HM) to over 99%. AMAISE increased all other accuracies by around a percentage point.

AMAISE + Centrifuge-M's classification time decreased to below that of Centrifuge-HM's as the percentage of host data in the test set increased. Specifically, as the host data in the test set increased from 1% to 99%, AMAISE + Centrifuge-M's classification time decreased linearly from 70 minutes to less than 20 minutes. Centrifuge-HM's classification time had the opposite trend -- as the host data in the test set increased from 1% to 99%, Centrifuge-HM's classification time increased linearly from 60 minutes to 70 minutes (Figure 4).

...

Across all test sets, AMAISE reduced the total storage requirements of Centrifuge-HM and Kraken2-HM by 18% and 14% and the peak memory usage by 17% and 14%, respectively.

Figure 4. Across samples that varied in terms of host percentage, the pipeline that included AMAISE consistently achieved a higher host accuracy while requiring less peak memory usage and remaining competitive with respect to classification time. Notably, AMAISE + Centrifuge-M's classification time decreased as the percentage of host data in the test set increased, while Centrifuge-HM's classification time increased.

New content in Methods (pg 17-18, line number 463-468)

Our 5 test sets with varying host fractions were each composed of 1,000,000 Nanopore sequences with human host percentages varying across 1%, 25%, 50%, 75%, and 99%. The remaining sequences in each test contained bacteria and fungi in the microbial fraction. The bacterial and fungal sequences were equally distributed among 216 bacterial species and 11 fungal species. The length of all the sequences ranges between 82 bp and 803,815 bp and the median length is 4108 bp.

Reviewer 1.3. The size of test sets further do not represent current metagenomic data sizes. 1,000,013 paired-end Illumina sequences, 1.2 GB, can hardly be regarded as a true reflection of the amount of data generated per sample in a metagenomic study.

Response 1.3. We have increased all test sets in our manuscript to 16 GB except the real metagenomic test set, which was 3 GB.

New content in Methods (pg 16, line number 412-413)

All test sets were 16 GB except the real metagenomic test set which was 3 GB.

Reviewer 1.4. No tests or reporting of tests done using "real" metagenomic data is reported. There is a wealth of public metagenomic data available for testing. Why were no comparisons done using some publicly available data?

Response 1.4. In our revised submission, we have included an additional test set with "real" metagenomic data. AMAISE achieves comparable or better performance across all evaluation metrics compared to Kraken2, Centrifuge, and Minimap2. However, the performance gap between AMAISE and other methods is not as large compared to that obtained on the synthetic samples for two reasons. First, the real metagenomic dataset contains shorter sequences (the real metagenomic dataset's median length of 930 bp vs. the synthetic data's median sequence length of 4108 bp), which Kraken2, Centrifuge, and Minimap2 can classify more quickly. Second, the host and microbial data in the real metagenomic dataset are more distinct than in the synthetic dataset. There are more unique 11-mers, 13-mers, 15-mers and 17-mers in the host data in the real metagenomic dataset compared to the synthetic dataset. Thus, classifying the real metagenomic dataset is an easier task than classifying the synthetic metagenomic data. This highlights situations in which AMAISE shines: when the input sequences are long and the task is difficult. We have updated the manuscript with these findings as described below.

New content in Results (pg 5, line 105-107)

Finally, to understand how our approach applies to real samples, we consider a test set of human oral metagenomic samples publicly available at the NCBI SRA (SRR11547004, SRR11547007).

New content in Results (pg 6-7, line number 151-170)

To test the ability of AMAISE when applied to real metagenomic data, we evaluated AMAISE on a test set of publicly available human oral metagenomic samples. On these data, AMAISE achieved a slightly higher specificity (91% vs. 85%-89%) and comparable sensitivity and accuracy to other approaches. AMAISE ran in 6 minutes which was about the same time it took to run Centrifuge, but required less storage (0.003 GB vs. 1.5 GB) (Figure 3a).

We found that the real metagenomic data contained, on average, shorter reads than the synthetic data, with a median read length of around 900 bp compared to the 4000 bp median read length for the synthetic data (Figure 3b). We also found that the host and microbial fractions in this data were more distinct than the fractions in the synthetic data. The host fraction in the real metagenomic data had more unique 11-mers, 13-mers, 15-mers, and 17-mers than the host fraction in the synthetic data (Figure 3c).

a) Results on a real metagenomic test set

b) Length of real metagenomic test set vs. the synthetic test set

c) Percentage of unique host k-mers in the real metagenomic test set compared to the synthetic test set

Figure 3. a) On real metagenomic data, AMAISE achieved a slightly higher specificity than other methods and remained competitive with respect to accuracy, sensitivity, and classification time. b) The real metagenomic data, on average, contained shorter sequences than the synthetic data. c) The real metagenomic data contains more unique 11-mers, 13-mers, 15-mers, and 17-mers in the host fraction compared to the synthetic data.

New content in Methods (pg 18, line number 469-472)

Our real metagenomic test set contained 800,000 sequences from a set of human oral metagenomic samples publicly available at the NCBI SRA (SRR11547004, SRR11547007). The length of all sequences ranged from 475 bp to 128,756 bp with the median sequence length being 930 bp.

Reviewer 1.5. The two read sets were assembled from sequencing reads obtained from the NCBI SRA, and did not have any post-sequencing modifications (e.g., modifications introduced by read cleaning and quality control procedures).", "However, recall that the reads in our test set did not undergo quality control." - In general, researchers will do sequence quality control as the first part of any project/analysis. The test set as such does not reflect true research protocols or pipelines. This further touches on the point above where the test sets are artificial and no test were done on "real" metagenomic data sets. In the manuscript the poor accuracy of Bowtie2 using raw sequencing data is highlighted and the accuracy improved after quality filtering. It seems that due to the inclusion of Nanopore data you were hesitant to include an initial quality filtering step in the testing procedure.

Response 1.5. While conventional quality control steps are currently standard, they are too time-consuming and impractical for real-time diagnoses. We developed AMAISE such that it could be applied before quality control techniques in research protocols and pipelines. This is especially useful when analyzing samples with a large amount of host data relative to the microbial data. Instead of applying time consuming quality control steps to a significant number of host DNA sequences, AMAISE will remove those host sequences, allowing users to only need

to apply quality control steps to the small number of microbial sequences. We have clarified in the Discussion section.

New content in Discussion (pg 13, line number 314-321)

We developed AMAISE such that it could be applied before quality control techniques in research protocols and pipelines. Existing quality control measures are time-consuming and impractical for real-time applications (such as clinical diagnostics) especially when applied to a significant number of host sequences. The ability of AMAISE to improve the accuracy and efficiency of existing methods without pre-processing via quality control and reference indices is a major advantage as we strive to bring sequencing to the bedside. This property is critical in resource-limited environments (e.g., clinical microbiology laboratories or point-of-care applications).

Reviewer 1.6. "Though CNNs have been used for metagenomic analysis in the past [7, 21-26]" - Why were no tests done against other CNN host/contamination depletion tools/applications currently available? If there are other CNN host/contamination depletion tools/applications currently available, what makes AMAISE novel?

Response 1.6. We have added comparisons to existing CNN-based approaches, highlighting the novelty and advance of AMAISE over these existing approaches. We updated the Results, Discussion, and Methods as follows.

New content in Results (pg 5-6, line number 109-123)

We compared AMAISE to CNN based approaches from past literature [13] [14] [15] in terms of the ability to classify a synthetic test set with a human host percentage of 50% and bacteria and fungi in the microbial fraction. We focused on two weaknesses of previously proposed solutions: the assumption of a fixed input size; and not training on data from both reference genomes and sequencing technologies. In contrast, AMAISE assumes a variable input size; and was trained on sequences from a combination of reference genomes and Nanopore data. We isolate each of these aspects and determine how much each contributes to the speed and accuracy of AMAISE.

In contrast to solutions that pad sequences to the maximum length [14], AMAISE's classified sequences more quickly (18 minutes vs. 31 minutes). While solutions that truncate all sequences to be length 250 are also fast [13], AMAISE outperforms such an approach in terms of accuracy by 10%.

Training on only reference genomes [14] or only data from a sequencing technology [13,15], results in lower sensitivity (79% vs. 99%) and lower specificity (73% vs. 99%) respectively, compared to training on both reference genomes and Nanopore data.

New content in Discussion (pg 13, line number 337-343)

Though CNNs have been used for metagenomic analysis in the past, AMAISE differs from past work in two ways. First, it is designed to efficiently and accurately process sequences of variable length. Past work in classifying Nanopore sequences either pad all sequences to a long sequence length [14] or truncate all sequences to a short sequence length [13] [15]. However, such padding increases classification time, while truncation ignores important information

leading to worse accuracy. Second, AMAISE is trained on a combination of reference genomes and Nanopore data. This leads to improved classification performance over training on reference genomes alone [14]; or sequenced data alone [13] [15].

New content in Methods (pg 20, line number 545-551)

We compared AMAISE to the following baseline CNN-based approaches from past literature: a CNN that pads all inputs to the maximum length [14] and a CNN that truncates all inputs to the minimum length [13]. We compared AMAISE to the following baseline CNNs to test the benefit of training on sequences sampled from reference genomes and Nanopore data: a CNN that trains solely on sequences from reference genomes [14] and a CNN that trains solely on sequences from Nanopore data [13] [15].

Reviewer 1.7. "First, it is designed to efficiently and accurately process sequences of variable length." - I assume that the variable length portion refers to Nanopore data. Is AMAISE the only program available to handle Nanopore data? Is this where the novelty of AMAISE lies?

Response 1.7. Yes, the variable length portion refers to Nanopore data. While other programs exist for processing Nanopore data they either are not machine learning-based or if they are they assume a fixed input size (e.g., by truncating or padding the sequence) and do not train on a combination of sequences from reference genomes and Nanopore data. We clarify this in the "Comparison to other CNN-based approaches" subsection under the Results section (Response 1.6).

Reviewer 1.8. In point 7 and in the text training is implied on a combination of reference genomes. In the manuscript it is stated that "We chose those three because they include the most common organisms and were closest to humans in their evolutionary distance, for which microbiome research is performed". Human, mouse and pig genomes were used for the training but only human was included in the validation and testing. Why not the other 2 which are "the most common"?

Response 1.8. We have included all methods' performance on samples with a pig host and a mouse host. AMAISE achieves comparable or better performance compared to existing approaches on these data. We have updated our Supplementary, Results, and Methods section accordingly (see below)

New content in Results (pg 7, line number 173-176)

To ensure that AMAISE could classify samples with non-human sequences, we evaluated AMAISE on these samples in Supplementary Figure 1 and confirmed that it had a high (> 98%) accuracy on those sequences.

New content in Supplementary Text (pg 1, line number 2-9)

a) Sensitivity and classification time on test set with pig host

b) Sensitivity and classification time on test set with mouse host

c) Length of test set with human host, pig host, and mouse host.

d) Percentage of unique host k-mers in test set with human host, pig host, and mouse host.

Figure 1. a) On data with a pig host, AMAISE achieved comparable sensitivity with much lower classification time than Centrifuge. b) AMAISE was faster than Minimap2 and Centrifuge at classifying the data with a mouse host, and had a higher sensitivity than Kraken2 and Centrifuge on these data. c) The samples with pig and mouse hosts, on average, contained longer reads than the samples with a human host. d) The samples with pig and mouse hosts contain more unique k-mers in the host data compared to the samples with a human host.

New content in Methods (pg 18, line number 473-480)

Our two test sets with non-human hosts either contained a pig or mouse host. These test sets contained 1,000,000 Nanopore sequences each with bacteria and fungi in the microbial fraction. The bacterial and fungal sequences were equally distributed among 216 bacterial species and 11 fungal species. The length of the sequences in the pig test set ranged between 82 bp and 153,007 bp and the median length was 19,995 bp. The test set contained 99% host data and 1% microbial data. The length of the sequences in the mouse test set ranged between 82 bp and 204,115 bp and the median length was 4,688 bp. The test set contained 99% host data and 1% microbial data.

Reviewer 1.9. "1,000,013 paired-end Illumina sequences, 99% of which are from a human and 1% of which are equally distributed among 16 bacteria and 3 fungi. 1,017,000 Nanopore sequences, 99% of which are from a human and 1% are equally distributed among 216 bacteria and 11 fungi." - Both test sets did not include viruses. Why not?

Response 1.9. We have included test sets with viruses. On this test set, AMAISE continued to outperform existing approaches in terms of accuracy and sensitivity and remained competitive in terms of speed and memory usage. The one metric that changed was specificity. We have updated our Supplementary and Methods section accordingly. We show the updated figures in the Supplementary section below and the updated Methods section describing the new test set.

New content in Results (pg 7, line number 176-179)

To ensure that AMAISE could classify samples with microbes that are similar to humans, we evaluated AMAISE on a sample with viruses in the microbial fraction in Supplementary Figure 2 and confirmed that it had a comparable specificity (92%) on those sequences to Centrifuge.

New content in Results (pg 9, line number 224-226)

Notably, though AMAISE achieved lower specificity than Kraken2-H on the test set with viruses in the microbial fraction, the pipeline with AMAISE and Kraken2-M achieved the same microbial accuracy as Kraken2-HM.

New content in Supplementary Text (pg 2, line number 10-15)

Figure 2. The two metrics that changed when adding viruses to our test set were specificity and microbial accuracies. AMAISE achieved comparable specificity to Centrifuge, and lower specificity than Kraken2 and Minimap2 (left figure). However, when used in a pipeline with Kraken2, AMAISE achieved the same microbial accuracy as Kraken2 (right figure).

New content in Methods (pg 18, line number 481-485)

Our test set with viruses contained 1,000,000 Nanopore sequences with a human host and bacteria, fungi, and viruses in the microbial fraction. The bacterial, fungal, and viral sequences were equally distributed among 216 bacterial species, 11 fungal species, and 52 viral species. The length of all the sequences ranged between 82 bp and 803,815 bp and the median length was 4,102 bp. The test set contained 99% host data and 1% microbial data.

Reviewer 1.10. “This represents a 300-fold improvement in storage efficiency over common classifications or alignment baselines.” – Again, not compared to other CNN applications. AMAISE is never explored or compared against other metagenomic CNN algorithms or machine learning approaches currently available.

Response 1.10. In Response 1.6, we compared AMAISE to existing CNN-based approaches that did not use our novel training setup (training on variable length sequences sampled from both reference genomes and Nanopore data and testing on variable length sequences) and showed that AMAISE performed better than these existing CNN-based approaches.

Reviewer 1.11. “AMAISE can be used in a number of different ways. Here, we investigate two specific applications i) as a stand-alone host-depletion tools and ii) in combination with existing metagenomic classifiers.” – No explanation of the number of different ways. The 2 ways as in the text are the same thing. AMAISE does host depletion and not classification. Point 2 is just an extension of the first part where classification is done on a host depleted set. Obviously, the other programs will perform better if an initial host depletion was done already.

Response 1.11. We agree that AMAISE can only be used one way, and have updated our Results section accordingly.

New content in Results (pg 4, line number 80-81)

We evaluated AMAISE i) as a stand-alone host-depletion tool and ii) in combination with existing metagenomic classifiers.

Reviewer 1.12. “Reassuringly, we found that the model had learned an association between ‘CG’ and a microbial classification.” - In essence, this model discriminates host and non-host based on GC content of a sequence. The question arises if classifying a sequence as host or non-host based on GC is novel enough.

Response 1.12. We appreciate the suggestion to determine how reliant AMAISE is on discriminating by ‘CG’ content. To further investigate whether or not AMAISE was relying on ‘CG’ content alone, we conducted additional experiments in which we i) compared AMAISE to a model that solely makes decisions based on the amount of ‘CG’ content in a sequence, and ii) compared AMAISE to Kraken2, Centrifuge, and Minimap2 on host/microbial sequences with similar amounts of ‘CG’ content. Our results indicate that not only does AMAISE rely on signals other than 'CG' content, but it relies far less on 'CG' content than existing approaches. We have updated the Results, Discussion and Methods section accordingly.

New content in Results (pg 11-12, line number 279-299)

However, AMAISE does not just rely on ‘CG’ to make its classification decisions. Compared to a naive method that solely used the amount of ‘CG’ in a sequence to make its classification decision, AMAISE was significantly more accurate (99% vs. 85%). Furthermore, we found that k-mer based and alignment based methods rely on ‘CG’ content to a much larger extent. Specifically, across subsets of the data with different ‘CG’ percentages, accuracy, sensitivity, and specificity all varied more for Kraken2, Centrifuge, and Minimap2 compared to AMAISE (Figure 7a).

a) Performance across 'CG' percentages

b) Percentage of unique host k-mers in test sets with different percentages of 'CG'

Figure 7. a) The performance of AMAISE remains stable across 'CG' percentages (< 1%, between 1 and 2%, between 2 and 4%, between 4 and 10%, and above 10%). Kraken2, Centrifuge, and Minimap2's sensitivity drops below 50% when the 'CG' percentage is above 4%. b) As the percentage of 'CG' in sequences increases, the number of unique 11-mers, 13-mers, and 15-mers in the host data decreases.

New content in Discussion (pg 14, line number 354-363)

However, AMAISE is not solely making its classification decisions based on the 'CG' content in sequences. Our results indicate that not only does AMAISE rely on signals other than 'CG' content, but it relies less on 'CG' content than existing approaches. We found that AMAISE's sensitivity and specificity were more stable and accurate across changes in 'CG' percentage than Kraken2, Centrifuge, and Minimap2. Specifically, Kraken2, Centrifuge, and Minimap2 had lower sensitivities than AMAISE on test sets with larger percentages of 'CG.' We found that the test sets with larger percentages of 'CG' had less unique k-mers in the host data compared to the test sets with smaller percentages (Figure 7b). Again, AMAISE's strength compared to other methods is its ability to accurately classify test sets even where the host and microbial data have similar k-mers.

New content in Methods (pg 18, line number 486-489)

To obtain our test sets with varying amounts of 'CG', we stratified one of our test sets into equal-sized quintiles by 'CG' percentage. The sequences were binned based on whether they had < 1% 'CG', between 1 and 2% 'CG', between 2 and 4% 'CG', between 4 and 10% 'CG', and < 10% 'CG.'

Reviewer 1.13. "AMAISE is not without limitations. First, AMAISE had lower specificity than Kraken2-H, Centrifuge-H, and Bowtie2-H on Illumina reads." – Only effective on nanopore reads. The low specificity may be due to sequence length and GC based on sequence length. The short Illumina sequences do not provide a strong enough signal to detect a true negative.

Response 1.13. We agree. We have followed your suggestion and refocused our manuscript on classifying long-reads sequences (as outlined above in Response 1.1).

Reviewer 1.14. "Finally, AMAISE was slower than Kraken2-H on both test sets, and Centrifuge-H on Illumina reads." – Not really that effective.

Response 1.14. We agree. Because of its ineffectiveness on Illumina reads, we have removed experiments on Illumina reads, focusing entirely on Nanopore reads, and highlight how the approach works best applied to long-reads.

Reviewer 1.15. "However, a user can tune AMAISE's decision threshold to increase specificity to microbial reads if privacy protection is not necessary", "Depending on the risk tolerance, a user can tune AMAISE's decision threshold to increase its sensitivity to host DNA." - In both these sentences there a 2 mentions which are worrisome. i) It seems like the user will have to tweak/tune the application when working with Illumina data. Can does not be done by the authors and incorporated in the model? Are these parameters/tuning efforts based on the composition of a sample and will it differ across samples? ii) The mention of "privacy protection" and "Depending on the risk tolerance". This needs to clarified and is currently very ambiguous.

Response 1.15. (i) Given the consistent performance of AMAISE across a variety of Nanopore test sets, we expect that our predefined thresholds will work well for most use cases. We clarify this in the Discussion section of our manuscript while adding that users can still adjust the thresholds of AMAISE if desired. (ii) By "privacy protection", we mean that some users may want to remove as much host data from a sample as possible to protect the privacy of a host. The "risk" we are referring to in "risk tolerance" is the amount of host DNA a user is willing to keep in a sample. If a user has a low "risk tolerance", they will want to remove as much host DNA from a sample as possible, potentially at the expense of removing some microbial DNA from a sample, and thus may be interested in decreasing AMAISE's decision threshold, which will likely classify more of the input data as host. If a user has a high "risk tolerance", they are more willing to keep host DNA in a sample. We have removed both of these phrases from our Discussion section and have replaced them with more informative phrases about what privacy protection means and who may be interested in changing AMAISE's decision threshold.

New Content in Discussion (pg 14-15, line number 381-389)

Despite these limitations, AMAISE's ability to accurately classify a variety of test sets makes it a versatile tool with wide-ranging applications and a solid base on which to build. Beyond its use in metagenomic classification pipelines, AMAISE can be used as a stand-alone host-depletion tool to remove host DNA in a sample to protect the privacy of the host [26]. Removing human DNA at the source could eliminate unnecessary exposure to sensitive host genomic data. Given the consistent performance of the proposed approach across a variety of test sets, we expect that AMAISE's default decision threshold will work well for the vast majority of use cases. However, if desired, a user could decrease AMAISE's decision threshold to increase the amount of host DNA that AMAISE removes.

Reviewer 1.16. The bar plots are of poor quality and can surely be improved.

Response 1.16. Thank you for your feedback. We have worked to improve the quality of the figures throughout the manuscript.

Reviewer 2

Reviewer 2.1: This article is about a new method to separate host-microbiota reads in a metagenomic set. I found this idea profoundly interesting since I have faced this problem by analyzing microbiomes mixed with unknown host reads. This is an important issue, microbial ecology has followed a significant improvement with NGS technologies, but all this host-genome DNA drags the study of non-human host-microbial systems metagenomic sets. However, the authors focus on the importance of their technique for clinical data; thus, the host = human.

Response 2.1: We have included all methods' performance on samples with a pig host and a mouse host. AMAISE achieves comparable or better performance compared to existing approaches on these data. We have updated our Supplementary, Results, and Methods section accordingly (see below).

New content in Results (pg 7, line number 173-176)

To ensure that AMAISE could classify samples with non-human sequences, we evaluated AMAISE on these samples in Supplementary Figure 1 and confirmed that it had a high (> 98%) accuracy on those sequences.

New content in Supplementary Text (pg 1, line number 2-9)

e) Sensitivity and classification time on

f) Sensitivity and classification time on

test set with pig host

g) Length of test set with human host, pig host, and mouse host.

test set with mouse host

h) Percentage of unique host k-mers in test set with human host, pig host, and mouse host.

Figure 1. a) On data with a pig host, AMAISE achieved comparable sensitivity with much lower classification time than Centrifuge. b) AMAISE was faster than Minimap2 and Centrifuge at classifying the data with a mouse host, and had a higher sensitivity than Kraken2 and Centrifuge on these data. c) The samples with pig and mouse hosts, on average, contained longer reads than the samples with a human host. d) The samples with pig and mouse hosts contain more unique k-mers in the host data compared to the samples with a human host.

New content in Methods (pg 18, line number 473-480)

Our two test sets with non-human hosts either contained a pig or mouse host. These test sets contained 1,000,000 Nanopore sequences each with bacteria and fungi in the microbial fraction. The bacterial and fungal sequences were equally distributed among 216 bacterial species and 11 fungal species. The length of the sequences in the pig test set ranged between 82 bp and 153,007 bp and the median length was 19,995 bp. The test set contained 99% host data and 1% microbial data. The length of the sequences in the mouse test set ranged between 82 bp and 204,115 bp and the median length was 4,688 bp. The test set contained 99% host data and 1% microbial data.

Reviewer 2.2: The authors claim AMAISE is more effective than other kraken2, minimap2, and bowtie2. AMAISE advantage is it is reference-free and fast. Nevertheless, the host reference genome known I do not see the point of adding a new bioinformatic tool, kraken2, can do the job.

Response 2.2: Due to AMAISE's low storage footprint, when applied before Kraken2, AMAISE results in a 14% decrease in memory and storage usage compared to using Kraken2 alone. As a stand-alone host depletion tool, AMAISE and Kraken2 perform comparably across evaluation metrics when the k-mers in the host and microbial data are distinct (80% of the 13-mers in the

host data are not present in the microbial data). However, when the host and microbial data become less distinct (15% of the 13-mers in the host data are not present in the microbial data, AMAISE consistently achieved a 15% better sensitivity with 0.3% of the storage requirements compared to Kraken2.

Reviewer 3

Reviewer 3.1. The AMAISE CNN is trained on various sequences drawn from a combination of reference genomes and reads. This gives AMAISE, as deployed here, visibility into genetic variation and sequencing error. This is in contrast to the k-mer-based approaches which, as deployed here, know only about reference genomes. On the one hand, we can consider this an inherent advantage of AMAISE. But on the other, there is nothing keeping the authors from also including genetic variation in the competing k-mer-based indexes. E.g. the authors could use more than one human reference genome in the k-mer indexes. Without such a comparison, it is hard to tell whether we're seeing an inherent advantage of AMAISE or an advantage of giving the classifier a "peek" at genetic variation. Note that some published index-based metagenomics classifiers are specifically geared toward using pan-genome indexes, and have been applied in host sequence removal scenario using multiple human genomes [doi:10.1016/j.isci.2021.102696].

Response 3.1. We agree that an advantage of AMAISE is that it is exposed to more genetic variation than the existing k-mer based approaches, and we can build pan-genome indices for k-mer based methods to give them the same advantage. However, we were unable to run Spumoni, the tool cited in the review, on our server with 225 GB of RAM – we received memory errors when computing the matching statistics. So instead, we ran Minimap2 with a pan genome index. We found that Minimap2's accuracy did not improve with this pan-genome index. We discuss this further in our Results and Supplementary text.

New content in Results (pg 6, line number 135-137)

We chose to only include the human genome assembly GRCh38.p13 in the reference indices of Kraken2-H, Centrifuge-H, and Minimap2-H because we found that Minimap2-H's accuracy decreased when using a pan genome index. We discuss this further in our Supplementary Text.

New content in Supplementary Text (pg 2, line number 16-26)

When using a pan-genome index to classify a read set with 99% host data, 1% bacterial and fungal data, Minimap2's accuracy did not improve. It actually decreased from 84% to 33%. While k-mer based indexes can include genetic variation via pan genomes, they cannot effectively include variation via large amounts of sequenced data from various sequencing technologies. A benefit of AMAISE compared to k-mer based approaches is that it can learn not just from reference genomes but from sequenced data that has not been aligned. Alignment inherently removes variation that can be caused by sequencing errors, making any method that learns from reference genomes less robust to sequencing errors. Thus AMAISE's advantage is not that it has a "peek" at general genetic variation, but that it can learn from sequenced reads that have not been aligned.

Reviewer 3.2. The authors explain the lower accuracy of Bowtie 2 by stating "the reads were not subjected to quality control, leaving adapter contamination or poor quality bases." Rather than use this tentative explanation, the authors should run Bowtie 2 with the --local option, which causes it to soft-clip low-quality or other poorly-matching bases from the extremes of the read.

Response 3.2. In response to Reviewer 1.1 that pointed out AMAISE's advantage compared to existing approaches was more apparent on Nanopore sequences, we have revised our manuscript to focus on classifying Nanopore sequences. Thus, we have removed all experiments with Bowtie2, since Bowtie2 is optimized to classify short reads. However, we ran Bowtie2 with the --local option. This increased Bowtie's accuracy and sensitivity, making its accuracy, sensitivity, and specificity above 99%. However, this also increased Bowtie2's classification time to almost twice that of AMAISE when the amount of host data in the test set is 99%.

Reviewer 3.3. A stated advantage of AMAISE is that it doesn't require a "memory-intensive" index data structure. But we read later that "In terms of peak memory usage, on both datasets, AMAISE was on par with Kraken2-H and Bowtie2-H, requiring 3.0 GB of RAM and 1.6 GB of video RAM (VRAM)." Not only does AMAISE's memory footprint seem comparable to methods that use a "memory-intensive" index (indeed, AMAISE's footprint is much larger than Centrifuge's), it also requires a specialized type of RAM (VRAM) that may not be as readily available to users. Is this really a good deal? The claims and discussion in the manuscript should be much more detailed and nuanced around this point.

Response 3.3. AMAISE's strength from a memory perspective is its ability to improve the peak memory and storage requirements of Kraken2 and Centrifuge as metagenomic classification tools. Given that a user has access to VRAM but a limited amount of RAM, AMAISE could be used to reduce the memory and storage requirements of existing metagenomic pipelines, allowing them to be used in RAM-limited environments (e.g. clinical microbiology laboratories or point-of-care applications). We clarify this and include AMAISE's usage of VRAM as a limitation of AMAISE in our discussion.

New content in Discussion (pg 14, line number 364-376)

AMAISE is not without limitations.

...

Second, AMAISE needs VRAM to speed up computation **and uses more RAM (has a larger memory footprint) than Centrifuge**. However, AMAISE uses less RAM than existing metagenomic classification methods and very little storage. This allows AMAISE to reduce the RAM and storage requirements of those methods by removing their need to store and reference host genomes. **This makes AMAISE a useful tool for a user that has access to VRAM but a limited amount of RAM and storage.**

Reviewer 3.4. Given that the explanation about what the model is learning revolves around the presence of CpGs, should we be concerned that the model will be differentially effective at classifying host (e.g. human) sequence from CpG islands? This would seem to be another relevant point for contrasting the CNN with index-based methods.

Response 3.4. To investigate whether AMAISE would be effective at classifying host sequences from CpG islands, we compared AMAISE to Kraken2, Centrifuge, and Minimap2 on host/microbial sequences with a large amount (>10%) of 'CG' content in them. AMAISE had a > 50% higher sensitivity (host accuracy) and comparable specificity (microbial accuracy) to existing approaches on this data (Figure 7 in Response 3.5). This indicates that AMAISE is much more effective at classifying host sequences with large amounts of 'CG' content than Kraken2, Centrifuge, and Minimap2. Thus, AMAISE will be effective at classifying host sequences from CpG islands.

Reviewer 3.5. Related to the previous point: do the authors expect that the AMAISE method will also work well in scenarios where the host genome and the target genome have a similar overall level of 'CG' content?

Response 3.5. To investigate whether AMAISE would be effective at classifying host sequences with similar overall levels of 'CG' content, we compared AMAISE to Kraken2, Centrifuge, and Minimap2 on host/microbial sequences with similar amounts of 'CG' content. AMAISE works well in scenarios where the host genome and target genome have similar overall levels of 'CG' content when classifying Nanopore reads. When classifying the test sets stratified by 'CG' content, AMAISE's sensitivity and specificity was above or comparable to other methods. We have updated our Results, Discussion, and Methods section accordingly.

New content in Results (pg 11-12, line number 279-299)

However, AMAISE does not just rely on 'CG' to make its classification decisions. Compared to a naive method that solely used the amount of 'CG' in a sequence to make its classification decision, AMAISE was significantly more accurate (99% vs. 85%). Furthermore, we found that k-mer based and alignment based methods rely on 'CG' content to a much larger extent. Specifically, across subsets of the data with different 'CG' percentages, accuracy, sensitivity, and specificity all varied more for Kraken2, Centrifuge, and Minimap2 compared to AMAISE (Figure 7a).

c) Performance across 'CG' percentages

d) Percentage of unique host k-mers in test sets with different percentages of 'CG'

Figure 7. a) The performance of AMAISE remains stable across 'CG' percentages (< 1%, between 1 and 2%, between 2 and 4%, between 4 and 10%, and above 10%). Kraken2, Centrifuge, and Minimap2's sensitivity drops below 50% when the 'CG' percentage is above 4%. b) As the percentage of 'CG' in sequences increases, the number of unique 11-mers, 13-mers, and 15-mers in the host data decreases.

New content in Discussion (pg 14, line number 354-363)

However, AMAISE is not solely making its classification decisions based on the 'CG' content in sequences. Our results indicate that not only does AMAISE rely on signals other than 'CG' content, but it relies less on 'CG' content than existing approaches. We found that AMAISE's sensitivity and specificity were more stable and accurate across changes in 'CG' percentage than Kraken2, Centrifuge, and Minimap2. Specifically, Kraken2, Centrifuge, and Minimap2 had lower sensitivities than AMAISE on test sets with larger percentages of 'CG.' We found that the test sets with larger percentages of 'CG' had less unique k-mers in the host data compared to the test sets with smaller percentages (Figure 7b). Again, AMAISE's strength compared to other methods is its ability to accurately classify test sets even where the host and microbial data have similar k-mers.

New content in Methods (pg 18, line number 486-489)

To obtain our test sets with varying amounts of 'CG', we stratified one of our test sets into equal-sized quintiles by 'CG' percentage. The sequences were binned based on whether they had < 1% 'CG', between 1 and 2% 'CG', between 2 and 4% 'CG', between 4 and 10% 'CG', and < 10% 'CG.'

Reviewer 3.6. Why is AMAISE's memory footprint so large? Do the CNN parameters take gigabytes of RAM?

Response 3.6.

We are assuming by memory footprint, the reviewer is referring to AMAISE's peak memory usage. AMAISE's CNN parameters take up very little storage (0.003 GB). Thus, AMAISE's peak memory usage is large not because of the CNN parameters. AMAISE's peak memory usage is large because the sequences classified by AMAISE need to be loaded into memory in order to be classified, and the computations that the CNN performs when classifying those sequences also need to be stored in memory during the classification process. The amount of memory AMAISE uses is directly proportional to the total length of the sequences AMAISE is classifying (the number of sequences multiplied by the length of each sequence). To decrease AMAISE's footprint, a user can decrease the total length of the sequences that AMAISE classifies, and we have detailed how to do so in the README that accompanies our code. The default total length makes AMAISE's memory footprint comparable to that of Kraken2. We have clarified this in our discussion.

New content in Discussion (pg 14, line number 370-379)

Second, AMAISE needs VRAM to speed up computation and uses more RAM (has a larger memory footprint) than Centrifuge. However, AMAISE uses less RAM than existing metagenomic classification methods and very little storage. This allows AMAISE to reduce the RAM and storage requirements of those methods by removing their need to store and reference host genomes. This makes AMAISE a useful tool for a user that has access to VRAM but a limited amount of RAM and storage. Furthermore, the amount of RAM and VRAM that AMAISE uses is a parameter that the user can change if they want AMAISE to be more memory efficient. The default parameter makes AMAISE's memory footprint comparable to that of Kraken2 and less than that of Minimap2.

Reviewer 3.7. The tools were evaluated on a test set that used a 99-to-1 mix of host versus non-host reads. Can the authors say anything about whether they expect AMAISE's relative performance to change much when a different mix is used? (E.g. 90-to-10 or 999-to-1?)

Response 3.7 We have added 5 additional Nanopore test sets that have the following levels of human contamination (1%, 25%, 50%, 75%, 99%). AMAISE's performance remained stable across test sets and was either comparable or better than that of the existing approaches. We have updated the Results and Methods sections as described below.

New content in Results (pg 5, line number 95-98)

In our samples, we first varied the percentage of reads of the sample that pertained to host. Specifically, we constructed 5 test sets with human host percentages (in percentage of number of reads) varying across 1%, 25%, 50%, 75%, and 99%, and each with bacteria and fungi in the microbial fraction.

New content in Results (pg 6-7, line number 138-184)

Across test sets with different host fractions, AMAISE achieved the highest sensitivity (>99%) compared to Kraken2-H, Centrifuge-H, and Minimap2-H. The existing method with the closest sensitivity, Centrifuge-H, achieved a sensitivity of only 87.5%. In addition, AMAISE achieved comparable or higher specificity and accuracy (> 98%) compared to existing methods. Across test sets with different host fractions, AMAISE ran in less than 20 minutes, on par with Kraken2-H and Minimap2-H, and between 10 and 50 minutes faster than Centrifuge-H (Figure 2).

Figure 2. Across samples that varied in terms of host percentage, the performance of AMAISE remained stable. AMAISE consistently achieved higher accuracy while requiring less storage and remaining competitive with respect to classification time and peak memory usage compared to other methods. Gains in accuracy and speed over Kraken2-H, Centrifuge-H, and Minimap2-H are most significant when the fraction of host samples is above 25%.

...

In terms of peak memory usage, on all datasets, AMAISE was on par with Kraken2-H, requiring 3.0 GB of RAM and 1.6 GB of video RAM (VRAM) compared to Kraken2-H's requirement of 4 GB of RAM. However, AMAISE required significantly less RAM than Minimap2-H. Finally, on all datasets, AMAISE required less than 0.3% of the storage required by the other existing approaches (Figure 2-3).

New content in Results (pg 8-9, line number 204-229)

Across test sets with different host fractions, the pipelines including AMAISE had higher host accuracies and microbial accuracies than the pipelines without AMAISE (Figure 4). AMAISE had the greatest impact on host sequence classification, improving the host accuracy of the pipelines from 85% (Centrifuge-HM) and 84% (Kraken2-HM) to over 99%. AMAISE increased all other accuracies by around a percentage point.

AMAISE + Centrifuge-M's classification time decreased to below that of Centrifuge-HM's as the percentage of host data in the test set increased. Specifically, as the host data in the test set increased from 1% to 99%, AMAISE + Centrifuge-M's classification time decreased linearly from 70 minutes to less than 20 minutes. Centrifuge-HM's classification time had the opposite trend

-- as the host data in the test set increased from 1% to 99%, Centrifuge-HM's classification time increased linearly from 60 minutes to 70 minutes (Figure 4).

...

Across all test sets, AMAISE reduced the total storage requirements of Centrifuge-HM and Kraken2-HM by 18% and 14% and the peak memory usage by 17% and 14%, respectively.

Figure 4. Across samples that varied in terms of host percentage, the pipeline that included AMAISE consistently achieved a higher host accuracy while requiring less peak memory usage and remaining competitive with respect to classification time. Notably, AMAISE + Centrifuge-M's classification time decreased as the percentage of host data in the test set increased, while Centrifuge-HM's classification time increased.

New content in Methods (pg 17-18, line number 463-468)

Our 5 test sets with varying host fractions were each composed of 1,000,000 Nanopore sequences with human host percentages varying across 1%, 25%, 50%, 75%, and 99%. The remaining sequences in each test contained bacteria and fungi in the microbial fraction. The bacterial and fungal sequences were equally distributed among 216 bacterial species and 11 fungal species. The length of all the sequences ranges between 82 bp and 803,815 bp and the median length is 4108 bp.

REVIEWERS' COMMENTS:

Reviewer #1 (Remarks to the Author):

Thank you for addressing all the concerns and comments. This was a huge task which you successfully concluded. Great job.

Reviewer #3 (Remarks to the Author):

The revision is thorough and addresses my main points. The manuscript is strong and ready to publish.